# Decreasing exciton dissociation rates for reduced voltage losses in organic solar cells

Hongbo Wu[1,7], Hao Lu[2,7], Yungui Li[3,7], Xin Zhou[3], Guanqing Zhou[4], Hailin Pan[1], Hanyu Wu [1], Xunda Feng [1], Feng Liu [4], Koen Vandewal [5], Wolfgang Tress [6], Zaifei Ma [1] ✉, Zhishan Bo [2] ✉ & Zheng Tang [1] ✉

Enhancing the device electroluminescence quantum efficiency ($EQE_{EL}$) is a critical factor in mitigating non-radiative voltage losses ($V_{NR}$) and further improving the performance of organic solar cells (OSCs). While the common understanding attributes $EQE_{EL}$ in OSCs to the dynamics of charge transfer (CT) states, persistent efforts to manipulate these decay dynamics have yielded limited results, with the $EQE_{EL}$ of high-efficiency OSCs typically remaining below $10^{-2}$%. This value is considerably lower than that observed in high efficiency inorganic photovoltaic devices. Here, we report that $EQE_{EL}$ is also influenced by the dissociation rate constant of singlet states ($k_{DS}$). Importantly, in contrast to the traditional belief that advocates maximizing $k_{DS}$ for superior photovoltaic quantum efficiency ($EQE_{PV}$), a controlled reduction in $k_{DS}$ is shown to enhance $EQE_{EL}$ without compromising $EQE_{PV}$. Consequently, a promising experimental approach to address the $V_{NR}$ challenge is proposed, resulting in a significant improvement in the performance of OSCs.

Continued development of organic donor and acceptor (D/A) photovoltaic materials[1–5] has resulted in power conversion efficiencies (*PCE*) of organic solar cells (OSCs) based on the bulk-heterojunction (BHJ) concept[6,7] approaching and exceeding 19%[8–11]. Both the photovoltaic external quantum efficiency ($EQE_{PV}$) and the fill-factor (*FF*) of state-of-the-art OSCs have already been improved to over 80%[12–15], approaching those of inorganic solar cells. However, the open-circuit voltage ($V_{OC}$) is still low, and increasing it is critically important for further improving the performance of OSCs.

$V_{OC}$ is limited in OSCs due to significant voltage losses associated with non-radiative recombination of charge carriers, referred to as non-radiative voltage losses ($V_{NR}$)[16,17]. The reported $V_{NR}$ for high-efficiency OSCs is as high as 0.20–0.30 V[18,19], while high-efficiency inorganic solar cells have a $V_{NR}$ of only 0.05 V[20,21]. In 2007, Rau demonstrated that $V_{NR}$ and $EQE_{EL}$ are related by Eq. (1)[22],

$$V_{NR} = -\frac{k_B T}{q}\ln(EQE_{EL}) \quad (1)$$

where $k_B$ is the Boltzmann constant, $q$ is the elementary charge, and $T$ is temperature. Therefore, a high $V_{NR}$ in OSCs goes hand-in-hand with a low electroluminescence quantum efficiency ($EQE_{EL}$), typically in the range of $10^{-5}$ % to $10^{-2}$ %.

For OSCs, $EQE_{EL}$ is primarily determined by the properties of the charge transfer (CT) state formed at the D/A interfaces in

[1]State Key Laboratory for Modification of Chemical Fibers and Polymer Materials, Center for Advanced Low-dimension Materials, College of Materials Science and Engineering, Donghua University, Shanghai 201620, PR China. [2]Key Laboratory of Energy Conversion and Storage Materials, College of Chemistry, Beijing Normal University, 100875 Beijing, PR China. [3]Max Planck Institute for Polymer Research, Ackermannweg 10, 55128 Mainz, Germany. [4]Frontiers Science Center for Transformative Molecules, School of Chemistry and Chemical Engineering, Shanghai Jiao Tong University, Shanghai 200240, PR China. [5]Instituut voor Materiaalonderzoek (IMO-IMOMEC), Hasselt University, Wetenschapspark 1, BE-3590 Diepenbeek, Belgium. [6]Institute of Computational Physics, Zurich University of Applied Sciences, Wildbachstr. 21, 8401 Winterthur, Switzerland. [7]These authors contributed equally: Hongbo Wu, Hao Lu, Yungui Li. ✉e-mail: mazaifei@dhu.edu.cn; zsbo@bnu.edu.cn; ztang@dhu.edu.cn

the BHJ active layer, given by[23,24]

$$EQE_{EL} = \frac{k_r^{CT}}{k_r^{CT} + k_{nr}^{CT}} = \frac{k_r^{CT}}{k_{CT}} \quad (2)$$

where $k_r^{CT}$ and $k_{nr}^{CT}$ are the radiative and non-radiative decay rate constants of the CT state, respectively, and $k_{CT}$ is the overall decay rate constant of the CT state. Due to strong vibrational coupling between the CT state and the ground state[25], $k_{nr}^{CT}$ of OSCs is high[23,25], being the main reason for the low $EQE_{EL}$.

Several attempts have been made to manipulate the decay dynamics of the CT state and thereby increase the $EQE_{EL}$ of OSCs. Eisner et al. [26] reported that increasing the degree of hybridization of the CT state and the singlet ($S_1$) state of the acceptor could result in increased $k_r^{CT}$, which in turn increased $EQE_{EL}$ and decreased $V_{NR}$, similar to that reported by Chen et al. [27] and Qian et al. [28] However, the increase in $k_r^{CT}$ also leads to an increased radiative voltage loss[29,30]. Ullbrich et al. reported that increasing the energy of the CT state ($E_{CT}$) could lead to weakened vibrational coupling between the CT state and the ground state[25], giving rise to reduced $k_{nr}^{CT}$, increased $EQE_{EL}$, and reduced $V_{NR}$, without affecting the radiative voltage losses. However, increasing $E_{CT}$ of OSCs requires the use of larger bandgap organic absorbers with a narrower absorption spectral range, which limits the short-circuit current density ($J_{SC}$) and consequently $PCE$.

In 2021, some of us demonstrated that increasing the spacing between D/A in the BHJ blend (DA spacing) could also lead to decreased $k_{nr}^{CT}$[31]. In that case, $EQE_{EL}$ was increased for the solar cell with increased DA spacing, and $V_{NR}$ was reduced without reducing $J_{SC}$. However, $V_{NR}$ remained high, over 0.3 V, and the $FF$ was limited for the solar cell with increased DA spacing. Thus, the performance of the solar cell was considerably worse than that of state-of-the-art OSCs. Recently, Gillett et al. demonstrated that the triplet states were an additional non-radiative decay channel for the CT states[32]. Increasing the degree of hybridization between the donor and acceptor molecules effectively prevented the decay of CT states via the triplet states, leading to increased $EQE_{EL}$. Nevertheless, despite considerable efforts that have been spent on the manipulation of the decay dynamics of CT state[33–38], $EQE_{EL}$ is still low for the high-efficiency OSCs. This low $EQE_{EL}$ problem, leading to high $V_{NR}$ and limited photovoltaic performance, seemed inevitable for OSCs[39].

In this work, we show that it is possible to increase the $EQE_{EL}$ of high-efficiency OSCs without compromising either $J_{SC}$ or $FF$, and without elevating radiative voltage losses. This is achieved by manipulating the dissociation dynamics of the $S_1$ state of the acceptor ($S_1^A$) rather than the decay dynamics of the CT state. Specifically, we reevaluate the dynamic processes of excited states in OSCs and illustrate that, beyond $k_r^{CT}$ or $k_{nr}^{CT}$, the rate constant governing the dissociation of $S_1^A$ ($k_{DS}$)—traditionally associated with charge carrier generation rate—likewise holds a significant role in determining the $EQE_{EL}$ of OSCs. Then, we demonstrate that, contrary to the prevailing belief that maximizing $k_{DS}$ is essential for achieving high $PCE$ in OSCs, a decrease in $k_{DS}$ can actually yield enhanced solar cell performance, provided that $k_{DS}$ remains above the threshold required for efficient exciton dissociation. In light of this, we propose an alternative avenue for research, one that centers on the manipulation of $k_{DS}$. This approach presents a promising solution to address the $V_{NR}$ issue and to overcome the $V_{OC}$ bottleneck in OSCs.

## Results

### Excited state dynamics in OSCs

Equation (2) was derived by assuming that the CT state was the sole decay channel for the excited states in OSCs[23]. However, this assumption does not hold for state-of-the-art OSCs based on non-fullerene acceptors (NFAs). In NFA OSCs, electroluminescence (EL) also includes $S_1^A$ emission, resulting from the back transfer of CT state

to the singlet state[26,40,41]. This introduces $S_1^A$ as an additional decay channel for free charge carrier recombination in NFA OSCs. A schematic illustration of the excited state dynamics for NFA OSCs under EL or $EQE_{EL}$ measurements is presented in Fig. 1a. When singlet population upon free charge carrier recombination becomes significant, the expression for the $EQE_{EL}$ for NFA solar cells becomes

$$EQE_{EL} = \frac{n_{S_1} k_r^{S_1} + n_{CT} k_r^{CT}}{n_{S_1} k_{S_1} + n_{CT} k_{CT}} \quad (3)$$

where $n_{CT}$ and $n_{S1}$ are the concentrations of the CT state and $S_1^A$, respectively, $k_{CT}$ and $k_{S1}$ are the decay rate constants of the CT state and $S_1^A$, respectively. Here, we assume that there is no decay of excited states via molecular triplet states.

In the case of OSCs operating as an LED in steady state, the generation rate of $S_1^A$ must be in equilibrium with the rate at which $S_1^A$ disappears. Then, we have

$$n_{CT} k_{BK} = n_{S_1} k_{S_1} + n_{S_1} k_{DS} \quad (4)$$

where $k_{DS}$ is the dissociation rate constant, i.e., the rate constant for the transition from $S_1^A$ to the CT state, $k_{BK}$ is the rate constant for the transition from the CT state to $S_1^A$. Combining Eqs. (3) and (4), we derive

$$EQE_{EL} = \frac{\eta_{S_1}}{1 + \left( \frac{1 + \frac{k_{DS}}{k_{S_1}}}{\frac{k_{BK}}{k_{CT}}} \right)} + \frac{\eta_{CT}}{1 + \left( \frac{\frac{k_{BK}}{k_{CT}}}{1 + \frac{k_{DS}}{k_{S_1}}} \right)} = \frac{\eta_{S_1}}{1 + \left( \frac{1 + r_{DS}}{r_{BK}} \right)} + \frac{\eta_{CT}}{1 + \left( \frac{r_{BK}}{1 + r_{DS}} \right)} \quad (5)$$

where $\eta_{CT}$ ($= \frac{k_r^{CT}}{k_{CT}}$) and $\eta_{S1}$ ($= \frac{k_r^{S_1}}{k_{S_1}}$) are the emission quantum efficiencies of the CT state and $S_1^A$, respectively, $r_{DS}$ is defined as the ratio between $k_{DS}$ and $k_{S1}$, and $r_{BK}$ is the ratio between $k_{BK}$ and $k_{CT}$. Note that $\eta_{S_1}$ is orders of magnitude higher than $\eta_{CT}$ for OSCs[27]. Therefore, from Eq. (5), one can derive that $EQE_{EL}$ increases when reducing $k_{DS}$, and the EL peak of the solar cell shifts from CT state emission to $S_1^A$ emission. Note that when the decay of excited states via molecular triplet states is involved, Eq. (3) requires modification, resulting in a new expression for the relationship between $EQE_{EL}$ and the rate constants, which differs slightly from Eq. (5). Nevertheless, the conclusion that $EQE_{EL}$ increases with the reduction of $k_{DS}$ remains valid. Further details on the derivation of Eq. (5), as well as the derivation of the expression with the decay via triplets included, can be found in Supplementary Note 1.

Similar to a derivation in the literature[42], by assuming that the built-in electric field is sufficiently large to dissociate CT states and extract free charge carriers, the internal quantum efficiency ($IQE$) of OSCs under short circuit can be expressed as (derivation of Eq. (6) is provided in the Supplementary Note 1)

$$IQE = \frac{\frac{k_{DS}}{k_{S_1}}}{1 + \frac{k_{DS}}{k_{S_1}}} = \frac{r_{DS}}{1 + r_{DS}} \quad (6)$$

Therefore, the maximum achievable $IQE$ decreases with reducing $k_{DS}$, but remains close to its maximal value as long as $k_{DS} \gg k_{S_1}$. Equations (5) and (6) express that for most of the typical OSCs with high $k_{DS}$, the CT state is the dominant emission channel and $EQE_{EL}$ is primarily determined by $\eta_{CT}$. Thus, $EQE_{EL}$ of these solar cells is low, $V_{NR}$ is high, and $IQE$ is also high. On the other hand, for OSCs with $k_{DS}$ being lower than $k_{S_1}$, $IQE$ is limited, but the dominant emission channel in these solar cells is $S_1^A$, giving rise to high $EQE_{EL}$ and low $V_{NR}$. There is clearly a trade-off between $IQE$ and $EQE_{EL}$, and thus, between $J_{SC}$ and $V_{OC}$ in BHJ OSCs, governed by $k_{DS}$.

To evaluate how $k_{DS}$ affects the overall performance of OSCs, we assume that $\eta_{S1}$ and $\eta_{CT}$ of a solar cell are 1% and $10^{-5}$%[27], respectively,

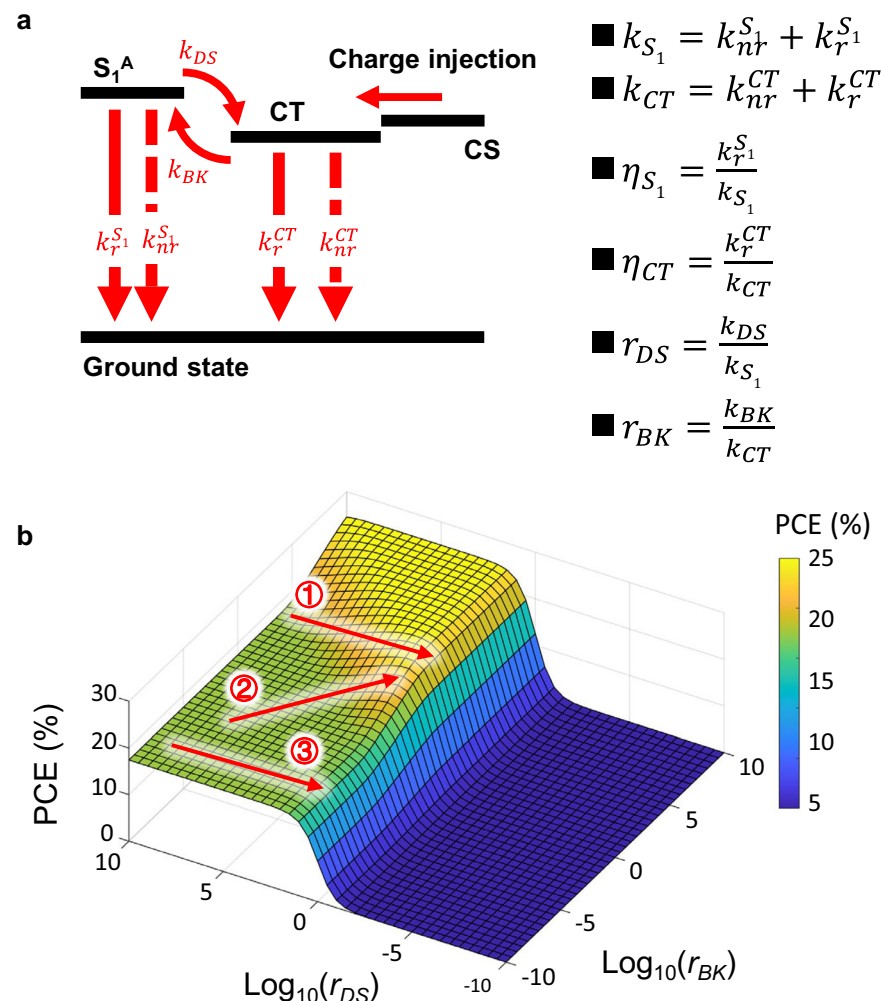

**Fig. 1 | Determining factors for the performance of non-fullerene organic solar cells. a** Dynamics of excited states in non-fullerene (NFA) organic solar cells (OSCs) under electroluminescence measurement. $S_1^A$, CT, and CS are abbreviations for acceptor singlet, charge transfer, and charge separated states, respectively. Note that for typical NFA OSCs, the energy of the singlet ($S_1$) state of the donor is much higher than that of the acceptor, and thus the concentration of the excited donor $S_1$ state is negligible at thermal equilibrium. **b** Power conversion efficiency (*PCE*) as a function of $r_{DS}$ and $r_{BK}$, calculated using Eq. (7). $\eta_{S_1}$ and $\eta_{CT}$ of the solar cell are assumed to be 1% and $10^{-5}$%, respectively. The radiative limit for the open-circuit voltage, the upper limit for the short-circuit current density, and the fill factor of the solar cell are assumed to be 1.10 V, 32 mA cm$^{-2}$, and 80%, respectively.

the radiative limit for $V_{OC}$ ($V_{OC}^{rad}$) of the solar cell is a constant, 1.10 V (implying that $k_{S_1}$ is lower than $k_{CT}$), the upper limit for $J_{SC}$ ($J_{SC}^{max}$) is 32 mA cm$^{-2}$, and the *FF* is 80%. Then, we relate the device *PCE* to *EQE_EL* and *IQE*, and thus, to $r_{DS}$ and $r_{BK}$, via

$$PCE = \frac{J_{SC} \cdot V_{OC} \cdot FF}{P_{in}} = \frac{J_{SC}^{max} \cdot IQE \cdot \left(V_{OC}^{rad} + \frac{kT}{q}\ln(EQE_{EL})\right) \cdot FF}{P_{in}}$$

$$= \frac{J_{SC}^{max} \cdot \frac{r_{DS}}{(1+r_{DS})} \cdot \left(V_{OC}^{rad} + \frac{kT}{q}\ln\left(\frac{\eta_{S_1}}{1+\left(\frac{1+r_{DS}}{r_{BK}}\right)} + \frac{\eta_{CT}}{1+\left(\frac{r_{BK}}{1+r_{DS}}\right)}\right)\right) \cdot FF}{P_{in}}$$

(7)

where $P_{in}$ is the incident light intensity (100 mW cm$^{-2}$). Accordingly, we calculate *PCE* as a function of $r_{DS}$ and $r_{BK}$, as shown in Fig. 1b.

For typical high-efficiency OSCs, the *IQE* is close to 100%, and $V_{NR}$ is also high, indicating that $r_{DS}$ is high and $r_{BK}$ is low. This signifies that the achievable *PCE* for typical OSCs is depicted by the greenish region in the lower-left corner of Fig. 1b. Therefore, three important conclusions regarding the impact of decreasing $k_{DS}$ and $r_{DS}$ on the performance of OSCs can be inferred from Fig. 1b:

1. When $r_{BK}$ is greater than a certain threshold, roughly $10^{-3}$ in the example above, the *PCE* of the solar cell will increase from 18% to over 25% with the decrease in $k_{DS}$, until $k_{DS}$ becomes excessively low, constraining *IQE* (indicated by the red arrow ① in Fig. 1b).
2. When $r_{BK}$ is lower than $10^{-3}$ and increases as $k_{DS}$ decreases, the *PCE* of the solar cell will also rise to over 25% with decreasing $k_{DS}$, until $k_{DS}$ becomes too low (indicated by the red arrow ② in Fig. 1b).
3. When $r_{BK}$ is lower than $10^{-3}$ but does not increase as $k_{DS}$ decreases, the *PCE* of the solar cell will not increase with decreasing $k_{DS}$ (indicated by the red arrow ③ in Fig. 1b).

Anticipating that $r_{BK}$ does not necessarily reduce with a reduction in $k_{DS}$, it follows that an intentional decrease in $k_{DS}$ within real-world OSCs could lead to an elevation in *PCE*.

### Fine-tuning $k_{DS}$ in OSCs using a dual-acceptor strategy

Practically, when considering binary OSCs, reducing $k_{DS}$ could require a reduction in the energy difference between the CT state and $S_1^A$ ($\Delta E_{CT}$)[43]. Achieving this requires precise adjustments to the energy levels of the frontier molecular orbitals of either the donor or acceptor material[44,45]. However, the task of such precise calibration presents challenges. Additionally, $k_{DS}$ depends on reorganization energy and

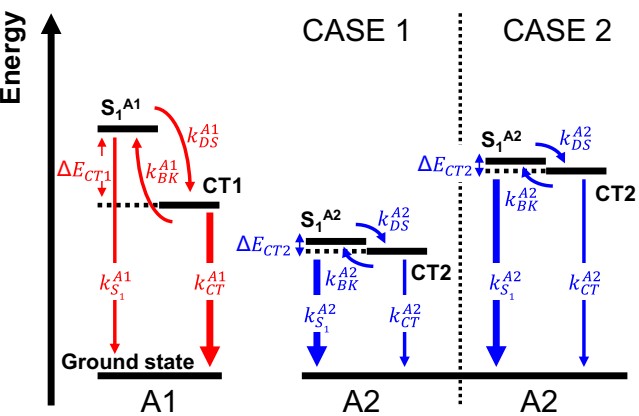

**Fig. 2 | Dynamics of excited states in the D/A1/A2 ternary solar cell.** A1 and A2 represent the primary and secondary acceptors used in the blend, respectively. CT1 and CT2 are the CT states formed at the D/A1 interfaces and the D/A2 interfaces, respectively. Because of the high $\Delta E_{CT1}$, $k_{DS}^{A1}$ is high, and due to the low $\Delta E_{CT2}$, $k_{DS}^{A2}$ is low. Thus, gradually increasing the A2 content in the D/A1/A2 ternary blend leads to gradually reduced $k_{DS}$ of the ternary blend system. Two different cases are illustrated, depending on the selection of A2, and the impact of the increased A2 content on the electroluminescence external quantum efficiency ($EQE_{EL}$) and the overall solar cell performance is anticipated to differ in these cases.

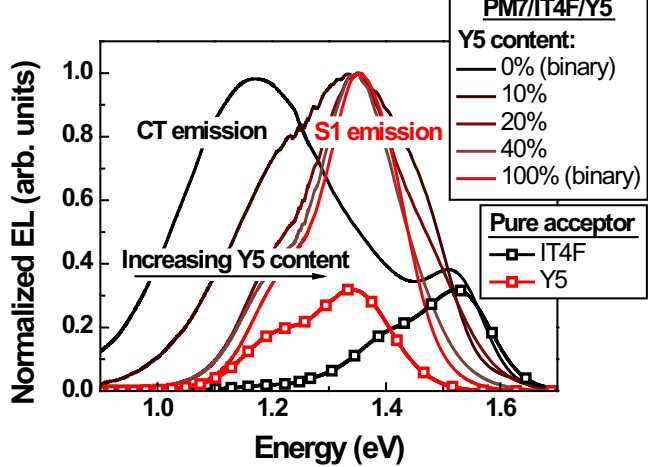

**Fig. 3 | Photophysical properties of PM7/IT4F and PM7/Y5.** Electroluminescence (EL) spectra of the solar cells based on the binary blends of PM7/IT4F and PM7/Y5, and the ternary blends of PM7/IT4F/Y5 with varying Y5 contents, and the EL spectra of the devices based on pure IT4F and pure Y5. Source data are provided as a Source Data file.

electronic coupling between the $S_1$ and CT states, both of which are even more challenging to adjust. Consequently, there is still a lack of an effective experimental strategy for optimizing the trade-off between $IQE$ and $EQE_{EL}$, through the regulation of $k_{DS}$.

In this work, we present a dual-acceptor strategy for the direct manipulation of $k_{DS}$ in OSCs. Differing from commonly employed ternary[46–50] or quaternary[51] strategies, which primarily focus on altering the microstructures of D/A blend films, our approach centers around modifying the energetics and dynamics of excited states within the blend films. Specifically, our active layer formulation for OSCs involves a common donor (D) combined with two different acceptor materials: a primary acceptor (A1) and a secondary acceptor (A2).

Within the D/A1/A2 ternary blend, the careful selection of A1 ensures that the energy of the CT state formed at the D/A1 interface (CT1) remains lower than the energy of the $S_1$ state of A1 ($S_1^{A1}$). Consequently, the difference between the energy of $S_1^{A1}$ and the CT state at the D/A1 interface ($\Delta E_{CT1}$) is anticipated to be substantial. Assuming the electronic coupling and reorganization energy of the blend of D/A1 are not significantly different from those of classic donor/acceptor blends used for organic solar cells, this high $\Delta E_{CT1}$ is expected to lead to a high dissociation rate constant of $S_1^{A1}$ ($k_{DS}^{A1}$) according to the Marcus theory[52,53]. Furthermore, our choice of A2 is designed to yield a CT state energy at the D/A2 interface (CT2) that closely aligns with the energy of the $S_1$ state of A2 ($S_1^{A2}$). This configuration results in a low $\Delta E_{CT}$ at the D/A2 interface ($\Delta E_{CT2}$), ideally approaching 0 eV, and consequently, a very low $k_{DS}^{A2}$. Hence, assuming a homogeneous distribution of the donor and acceptors in the D/A1/A2 ternary active layer, $k_{DS}$ for the ternary blend, expressed as the weighted average of $k_{DS}^{A1}$ and $k_{DS}^{A2}$, can be modulated by adjusting the A2 content, similar to what has been reported in the literature for ternary blends with two donors[54]: A gradual decrease of $k_{DS}$ is possible to achieve by progressively increasing the A2 content.

It should be noted that the influence of increased A2 content on $EQE_{EL}$ and the overall performance of the ternary solar cell is expected to diverge based on the selection of A2. Illustrated through the dynamic processes of excited states in the dual acceptor solar cell (Fig. 2), two distinct scenarios emerge. In both situations, $k_{DS}$ of the solar cell reduces with increased A2 content. In the first scenario, A2 possesses an $S_1$ state energy lower than the $E_{CT}$ of the D/A1 blend. Here, increasing A2 content results in lowered $E_{CT}$ and subsequently elevated $k_{CT}$ of the ternary blend[55,56], leading to a constrained increase in $EQE_{EL}$

in accordance with Eq. (5). In the second scenario, A2 has an $S_1$ state energy close to or higher than the $E_{CT}$ of the D/A1 blend. In this case, increasing A2 content has a limited impact on $E_{CT}$ and $k_{CT}$ of the ternary blend, thus resulting in an increased $EQE_{EL}$.

## Impact of reduced $k_{DS}$ on the performance of OSCs

To examine the effectiveness of the dual-acceptor strategy in modulating the $k_{DS}$ of organic blends, solar cells were constructed using the conventional device architecture of ITO/PEDOT:PSS/active layer/PFN-Br/Ag. Initially, we employed IT4F as the primary acceptor, Y5 as the secondary acceptor, and PM7 as the donor[57–59]. The chemical structures of the active materials are provided in Supplementary Fig. 1.

We selected PM7/IT4F as our D/A1 blend because the EL spectrum of the PM7/IT4F binary solar cell was predominantly characterized by CT state emission (Fig. 3). This observation strongly suggests a high $\Delta E_{CT}$ for the PM7/IT4F blend, which we estimate to be 0.12 eV (Supplementary Fig. 2). Therefore, $k_{DS}$ for the PM7/IT4F blend is expected to be high, which can also be seen from femtosecond transient absorption (fs-TA) measurements.

Specifically, a weak pump excitation at 750 nm (1.7 µJ cm⁻²) was selected for the fs-TA measurements to exclusively excite the $S_1$ state of IT4F (UV-vis absorption spectra of PM7 and IT4F are provided in Supplementary Fig. 3). In this case, the only charge transfer process in the PM7/IT4F blend system during the fs-TA measurements involved hole transfer from IT4F to PM7. As a result, the negative transient absorption signals (expressed as ΔA, Supplementary Fig. 4) were detected in the sub-650 nm region and the 700–800 nm region of the fs-TA spectrum, which are attributed to the ground state bleaching (GSB) of the donor PM7 and acceptor IT4F, respectively. Because these spectral response coincide with the steady state absorption spectra of PM7 and IT4F (Supplementary Fig. 3). Consistent with the findings reported in the literature[60], the GSB signals of the donor (at 630 nm) persisted for a relative extended duration, thus they can be used to represent the charge generation kinetics, i.e., the transfer of holes from the acceptor to the donor. Therefore, the time evolution of the TA signals at 630 nm is fitted using the sum of two exponential functions and an instrument response function (IRF), giving rise to a rise time constant of ≈8 ps. Importantly, the GSB signal of PM7 is observed quickly after photoexcitation, which implies an ultra-fast hole transfer from the acceptor to the donor, corresponding to a high $k_{DS}$ value, thanks to the high $\Delta E_{CT}$ for the PM7/IT4F blend.

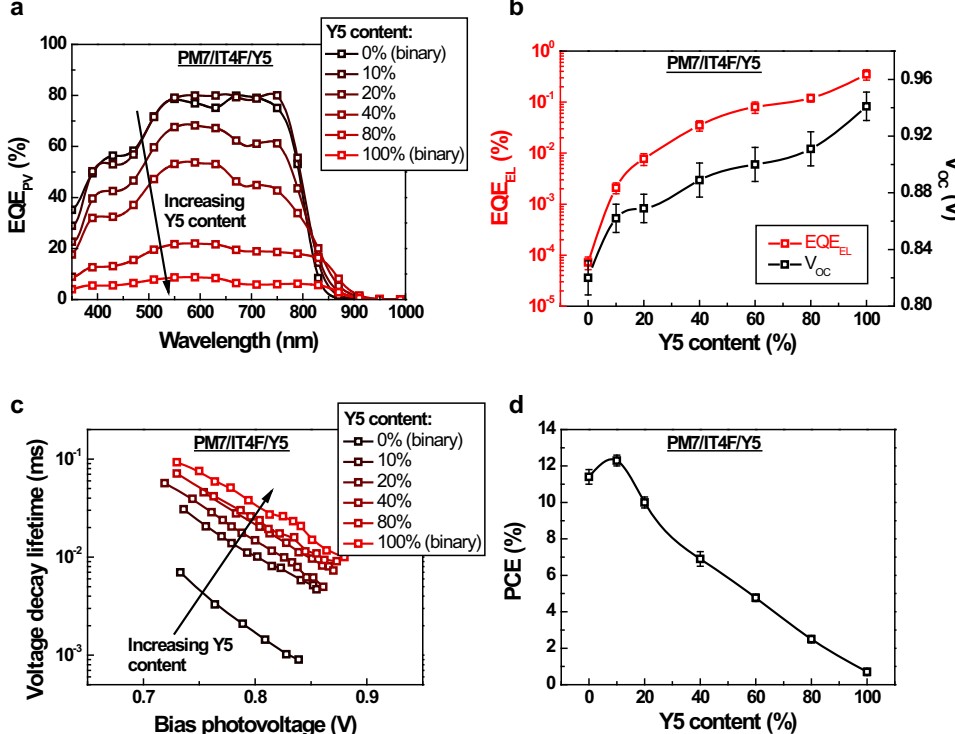

**Fig. 4 | Performance of PM7/IT4F/Y5 ternary solar cells. a** Photovoltaic external quantum efficiency ($EQE_{PV}$) of the PM7/IT4F/Y5 ternary solar cells with different Y5 contents. **b** Electroluminescence external quantum efficiency ($EQE_{EL}$) and open-circuit voltage ($V_{OC}$) of the PM7/IT4F/Y5 ternary solar cells with different Y5 contents. The $EQE_{EL}$ values are determined using an injection current density equal to the short-circuit current density ($J_{SC}$) of the device. **c** Transient photovoltage (TPV) decay lifetime of the PM7/IT4F/Y5 ternary solar cells with different Y5 contents (see Supplementary Note 3 for details). **d** Power conversion efficiency ($PCE$) of the PM7/IT4F/Y5 ternary solar cells with different Y5 contents. The error bars in **b** and **d** represent the highest, lowest, and average values from multiple devices. Source data are provided as a Source Data file.

We opted for Y5 as the secondary acceptor A2 due to the emission solely from the $S_1$ state of Y5 in the EL spectrum of the PM7/Y5 binary solar cell (Fig. 3). This observation indicates a low $\Delta E_{CT}$ within the PM7/Y5 blend, and consequently, a low anticipated dissociation rate constant. We conducted fs-TA measurements for the blend film of PM7/Y5 and observed negative signals in the 650-850 nm region (Supplementary Fig. 4), assigned to the GSB of Y5 (steady-state absorption spectra of Y5 shown in Supplementary Fig. 3). However, unlike the PM7/IT4F blend, GSB signals from the donor PM7 are hardly observed in the fs-TA spectra for the PM7/Y5 blend.

In fact, the TA spectra for the PM7/Y5 system closely resemble those for the neat Y5 film, as shown in Supplementary Fig. 5. Moreover, a comparison of the GSB signals for Y5 in the neat film and those for the PM7/Y5 blend reveals very similar decay lifetimes (≈140 ps). These experimental TA features indicate that in the PM7/Y5 system, the intrinsic kinetics of Y5 dominate the photophysical process, while the contribution of the PM7 GSB signal is minimal. This could be due to either the absence of the dissociation of acceptor excitons (no transfer of holes from Y5 to PM7), or, alternatively, the holes, after transferring from Y5 to PM7, could quickly return from PM7 to Y5, owing to the extremely lower $\Delta E_{CT}$. In both cases, the hole transfer rate is severely limited. Therefore, the $k_{BK}$ of the PM7/Y5 blend, with a lower $\Delta E_{CT}$, is higher compared to that of PM7/IT4F, corresponding to a lower $k_{DS}$ for the PM7/Y5 blend. Similar results have been reported for organic donor/acceptor systems, where a reduced $\Delta E_{CT}$ leads to a reduction in hole transfer rate by over two orders of magnitudes[53].

Since the fs-TA results indicated a lower $k_{DS}$ in PM7/Y5 compared to PM7/IT4F, it is anticipated that the $k_{DS}$ will decrease as the Y5 content increases within the ternary blend of PM7/IT4F/Y5. This is confirmed by the electric field dependent steady state PL measurements, as detailed in Supplementary Note 2. Furthermore, given that the $S_1$

state energy of Y5 (1.43 eV, Supplementary Fig. 3) closely approximates the $E_{CT}$ of the PM7/IT4F blend (1.46 eV), the excited state dynamics within the PM7/IT4F/Y5 ternary blend align with the scenario depicted as case 2 in Fig. 2. Hence, the decreased $k_{DS}$ also prompts a shift in the primary decay pathway within ternary solar cells, transitioning from the CT state to the $S_1$ state of Y5 as the Y5 content increases. This transition is indicated by the alteration of the predominant emission peak in the EL spectrum (Fig. 3) from the PM7/IT4F CT state (1.2 eV) to the Y5 $S_1$ state (1.35 eV). Consequently, the device $EQE_{EL}$ is significantly increased, as shown in Fig. 4b: For the ternary solar cell with a Y5 content of 10%, i.e., the ratio between the weight of Y5 and the total weight of the acceptors, $EQE_{EL}$ is increased by over an order of magnitude compared to the PM7/IT4F binary solar cell. Thus, $V_{NR}$ of the ternary solar cell is reduced from 0.35 to 0.27 V (Table 1).

Transient photovoltage (TPV) measurements were also performed on the PM7/IT4F/Y5 ternary solar cells to evaluate the impact of increasing Y5 content on the lifetimes of charge carriers. The results are presented in Fig. 4c. Clearly, as the Y5 content increases, the photovoltage lifetime increases, indicating an extended lifespan of charge carriers. As a result, the increased $EQE_{EL}$ and reduced $V_{NR}$ contribute to an overall reduction in voltage loss and an increase in $V_{OC}$, as demonstrated in Fig. 4b: The $V_{OC}$ of the solar cell increases from 0.82 V to 0.87 V as the Y5 content rises from 0% to 10%. It is important to note that the substantial increase in $V_{OC}$ underscores the distinct nature of the dual-acceptor strategy proposed herein for manipulating $k_{DS}$, diverging from conventional ternary strategies. Since without accounting for the influence of $k_{DS}$, the addition of a third component with an $S_1$ state energy lower than the primary D/A system would conventionally lead to a reduction in $V_{OC}$ for ternary OSCs[61–67].

It should be noted that the addition of Y5 to the binary blend of PM7/IT4F could lead to a change in the morphology of the active layer,

**Table 1 | Photovoltaic performance of the PM7/IT4F/Y5 solar cells**

| Y5 content | $V_{OC}$ (V) | $J_{SC}$ (mA cm$^{-2}$) | FF (%) | PCE (%) | $EQE_{EL}$ (%) | $V_{NR}$ (eV) |
|---|---|---|---|---|---|---|
| 0% | 0.827 (0.820 ± 0.012) | 19.2 (19.1 ± 0.3) | 74.4 (74.0 ± 1.2) | 11.8 (11.4 ± 0.4) | $7.2 \times 10^{-5}$ | 0.354 |
| 10% | 0.869 (0.862 ± 0.010) | 19.7 (19.5 ± 0.4) | 73.5 (73.1 ± 1.4) | 12.6 (12.3 ± 0.3) | $2.1 \times 10^{-3}$ | 0.269 |
| 20% | 0.876 (0.869 ± 0.011) | 16.2 (16.0 ± 0.4) | 72.6 (72.5 ± 1.3) | 10.3 (10.0 ± 0.3) | $7.7 \times 10^{-3}$ | 0.237 |
| 40% | 0.895 (0.889 ± 0.012) | 12.7 (12.6 ± 0.5) | 64.2 (64.0 ± 0.9) | 7.3 (6.9 ± 0.4) | $3.5 \times 10^{-2}$ | 0.199 |
| 80% | 0.918 (0.911 ± 0.012) | 5.8 (5.7 ± 0.3) | 48.8 (48.6 ± 1.5) | 2.7 (2.5 ± 0.2) | $1.2 \times 10^{-1}$ | 0.168 |
| 100% | 0.949 (0.941 ± 0.010) | 2.1 (2.0 ± 0.3) | 44.8 (44.5 ± 1.2) | 0.9 (0.7 ± 0.2) | $3.5 \times 10^{-1}$ | 0.141 |

Representative performance parameters of the solar cells based on PM7/IT4F/Y5, with different Y5 contents. The *J-V* curves of the devices are provided in Supplementary Fig. 8. The statistic results are obtained from 8 individual devices. The $EQE_{EL}$ values are determined using an injection current density equal to the $J_{SC}$ of the device.

potentially resulting in a reduction in the reorganization energy of the active layer. This, in turn, could contribute additionally to the increase in $EQE_{EL}$[23,24]. Therefore, we measured the EL and sensitive $EQE_{PV}$ spectra of the solar cell based on PM7/IT4F, PM7/IT4F/Y5, and PM7/Y5 (Supplementary Fig. 6) and determined the reorganization energy through Gaussian fitting to the tail of the $EQE_{PV}$ spectra. We found that the reorganization energy is approximately 0.32 eV for the PM7/IT4F binary blend, and it slightly decreases to 0.28 eV in the ternary blend with a Y5 content of 20%. This suggests that the introduction of Y5, while it may cause changes in the morphology of the active layer, does not significantly decrease the reorganization energy. Therefore, the increased $EQE_{EL}$ should not be ascribed to a morphological reason.

Next, we explore the implications of decreasing $k_{DS}$ on *IQE* and $EQE_{PV}$ of the PM7/IT4F/Y5 ternary solar cell. As illustrated in Fig. 4a, our initial observation highlights a significant difference between the $EQE_{PV}$ of the binary PM7/IT4F (over 80%) and PM7/Y5 solar cells (below 10%). The markedly low $EQE_{PV}$ for the PM7/Y5 solar cell is a foreseeable outcome, attributed to the lower $\Delta E_{CT}$ and $k_{DS}$ within the PM7/Y5 blend. This leads to a restricted *IQE* of the binary solar cell, a hypothesis that gains support from the steady state PL quenching measurements, as further elucidated in Supplementary Fig. 7. Nevertheless, the $EQE_{PV}$ of the PM7/IT4F solar cell remains unaffected by the introduction of a small quantity of Y5: The peak $EQE_{PV}$ of the ternary solar cell with a Y5 content of 10% is as high as that of the PM7/IT4F binary solar cell (≈80%). It is only when the Y5 content surpasses 10% that a distinct reduction in the peak $EQE_{PV}$ becomes evident. This pattern suggests that the $k_{DS}$ within the PM7/IT4F binary blend substantially surpasses the threshold required to achieve an *IQE* close to 100%. Consequently, a controlled reduction in $k_{DS}$ for high quantum efficiency OSCs does not necessarily lead to a direct decrease in *IQE*, as predicted by Eq. (6).

Since the *FF* of the ternary solar cell is hardly affected by the addition of a small amount of the secondary acceptor (Table 1), *PCE* of the solar cell could indeed be increased using the dual-acceptor strategy: The *PCE* of the solar cell is increased from 11.8% to 12.6%, when the Y5 content is increased from 0% to 10% (Fig. 4d). The basic photovoltaic performance parameters of the solar cells with different Y5 contents are summarized in Table 1, and the current density-voltage (*J-V*) curves are provided in Supplementary Fig. 8.

We have also employed the dual-acceptor strategy for solar cells based on the blend of PM7/IT4F as the primary D/A1 system, and Y5OD as A2, as well as solar cells based on PM6/fullerene as the primary D/A system, and an non-fullerene acceptor SM16 as A2. In all cases, we realized increased $EQE_{EL}$, reduced overall voltage loss, and improved solar cell performance by a controlled decrease in $k_{DS}$, achieved by gradually increasing the A2 content. More details are provided in Supplementary Note 4 and 5. The outcomes derived from the study of these ternary solar cells all demonstrate the potential for enhancing the performance of OSCs through the reduction of $k_{DS}$. Nonetheless, it is important to underline that these material combinations were selected as model systems, because: 1) The primary D/A1 system possesses high $k_{DS}$, giving rise to high *IQE*, while the secondary

D/A2 system has low $k_{DS}$. 2) The introduction of A2 content is not anticipated to significantly influence $k_{S_1}$ and $\eta_{S_1}$ within the primary D/A1 blend, because of the comparable $S_1$ state lifetime values between A1 and A2 (the lifetime values determined from fs-TA are 40 and 140 ps for IT4F and Y5, respectively, see Supplementary Fig. 5). 3) The emission efficiencies of the $S_1$ states of A1 and A2 also appear similar (the $EQE_{EL}$ of the solar cells based on pure IT4F and pure Y5 are 0.2% and 0.3%, respectively, see Supplementary Fig. 9).

The well-understood electronic properties of these material combinations allow us to have confidence in the effectiveness of the dual-acceptor strategy in modulating $k_{DS}$, as well as in the deduction concerning the pivotal role of $k_{DS}$ in shaping the trade-off between $EQE_{EL}$ and *IQE*, and improving overall solar cell performance. However, the decrease in *IQE* is significant, while the increase in $EQE_{EL}$ is less pronounced for these solar cells when the A2 content exceeds 10%. This phenomenon could be attributed to the substantial molecular structural differences between the primary acceptor A1 and the secondary acceptor A2, leading to different stacking characteristics of the donor or the acceptor molecules in the ternary active layer with increasing A2 content (see Supplementary Fig. 10). As a result, the dual-acceptor strategy results in a limited enhancement in *PCE* for solar cells based on the PM7/IT4F/Y5, PM7/IT4F/Y5OD, or PM6/fullerene/SM16. This challenge will be addressed in the subsequent section of this article.

## Improving the performance of high-efficiency OSCs

To further explore the advantages of the dual-acceptor strategy in enhancing the performance of high-efficiency OSCs and to establish the general applicability of the strategy, we employed D18[68] and EC9[69] as the primary D/A1 system. For A2, we utilized SM16[70], which bears a molecular structure similar to EC9. The chemical structures of these active materials are displayed in Supplementary Fig. 1.

Before constructing the ternary devices, we first studied the binary solar cells based on D18/EC9 and D18/SM16. In the D18/EC9 binary solar cell, the EL spectrum predominantly features an emission peak corresponding to the $S_1$ state of EC9 (Fig. 5a), implying a low $\Delta E_{CT}$ for the solar cell and a reduced $k_{DS}$ in comparison to the PM7/IT4F binary solar cell. However, the $k_{DS}$ within the D18/EC9 binary solar cell remains sufficiently high to facilitate efficient exciton dissociation, which results in a high *IQE* and a peak $EQE_{PV}$ of over 80% (Fig. 5b). The $EQE_{EL}$ of the solar cell is constrained to around 0.02% (Fig. 5c), consequently leading to a high $V_{NR}$, exceeding 0.2 V (Table 2).

On the other hand, the D18/SM16 binary solar cell exhibits a predominant emission peak in the EL spectrum also stemming from the $S_1$ state of the acceptor (Fig. 5a). This signifies a low $\Delta E_{CT}$. However, the $k_{DS}$ within the D18/SM16 binary solar cell is expected to be excessively low, significantly impeding the efficiency of exciton dissociation, as validated by PL quenching measurements (Supplementary Fig. 11). Consequently, the *IQE* remains low, and the peak $EQE_{PV}$ is confined to less than 5% (Fig. 5b). Owing to the markedly low $k_{DS}$, the $EQE_{EL}$ of the D18/SM16 binary solar cell is high, approximately 0.7% (Fig. 5c). This is paralleled by a low $V_{NR}$ of merely 0.12 V (Table 2).

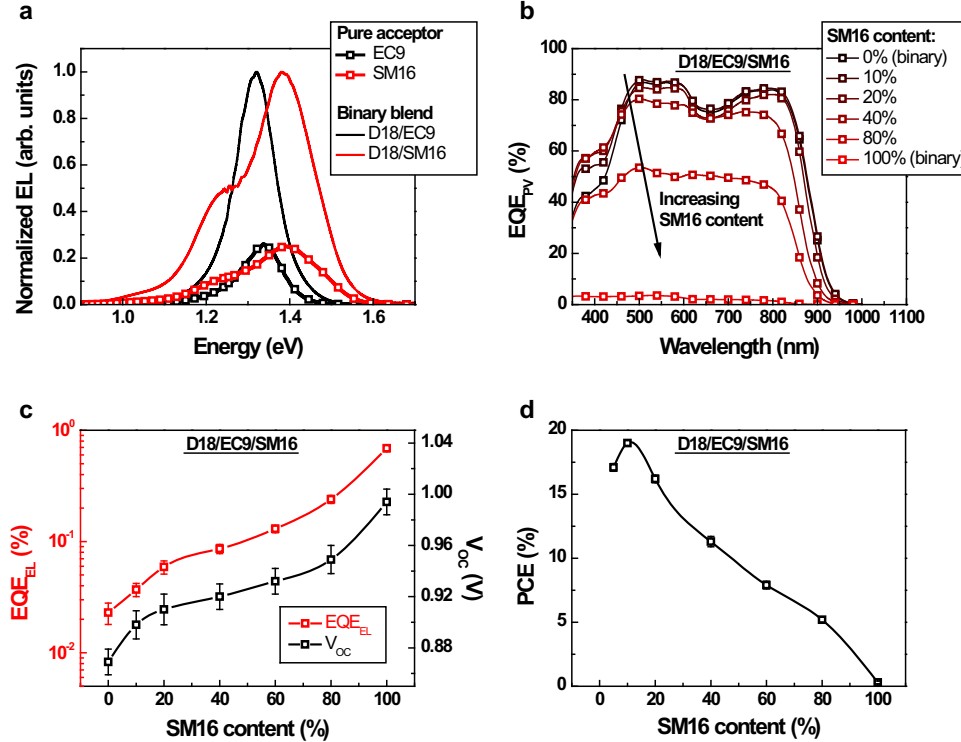

**Fig. 5 | Performance of the solar cells based on D18/EC9/SM16.**
**a** Electroluminescence (EL) spectra of the solar cells based on the binary blends of D18/EC9 and D18/SM16, and the EL spectra of the devices based on pure EC9 and pure SM16, **b** Photovoltaic external quantum efficiency ($EQE_{PV}$) of the solar cells based on D18/EC9/SM16 with different SM16 contents. **c** Electroluminescence external quantum efficiency ($EQE_{EL}$) and open-circuit voltage ($V_{OC}$) of the D18/EC9/SM16 ternary solar cells with different SM16 contents. **d** Power conversion efficiency (*PCE*) of the D18/EC9/SM16 ternary solar cells with different SM16 contents. The error bars in **c** and **d** represent the highest, lowest, and average values from multiple devices. Source data are provided as a Source Data file.

**Table 2 | Photovoltaic performance of the D18/EC9/SM16 solar cells**

| SM16 content | $V_{OC}$ (V) | $J_{SC}$ (mA cm$^{-2}$) | FF (%) | PCE (%) | $EQE_{EL}$ (%) | $V_{NR}$ (eV) |
|---|---|---|---|---|---|---|
| 0% | 0.869 (0.865 ± 0.010) | 26.7 (26.5 ± 0.2) | 74.9 (74.5 ± 1.1) | 17.4 (17.1 ± 0.3) | $2.3 \times 10^{-2}$ | 0.210 |
| 10% | 0.906 (0.898 ± 0.011) | 27.0 (26.8 ± 0.2) | 78.5 (78.0 ± 1.2) | 19.2 (19.0 ± 0.2) | $3.7 \times 10^{-2}$ | 0.198 |
| 20% | 0.914 (0.910 ± 0.012) | 27.1 (26.8 ± 0.3) | 66.6 (66.3 ± 1.3) | 16.5 (16.2 ± 0.3) | $5.9 \times 10^{-2}$ | 0.186 |
| 40% | 0.925 (0.920 ± 0.010) | 22.9 (22.5 ± 0.4) | 55.5 (55.0 ± 1.3) | 11.7 (11.3 ± 0.4) | $8.6 \times 10^{-2}$ | 0.176 |
| 80% | 0.956 (0.949 ± 0.011) | 14.4 (14.0 ± 0.5) | 39.6 (39.2 ± 1.4) | 5.4 (5.2 ± 0.2) | $2.4 \times 10^{-1}$ | 0.151 |
| 100% | 1.000 (0.994 ± 0.010) | 0.8 (0.7 ± 0.1) | 50.6 (50.2 ± 1.0) | 0.4 (0.3 ± 0.1) | $6.9 \times 10^{-1}$ | 0.124 |

Representative performance parameters of the solar cells based on D18/EC9/SM16, with different SM16 contents. The *J-V* curves of the devices are provided in Supplementary Fig. 16. The statistic results are obtained from 8 individual devices. The $EQE_{EL}$ values are determined using an injection current density equal to the $J_{SC}$ of the device.

It should also be noted that the $S_1$ state energy of SM16 is comparable to that of EC9 (Supplementary Fig. 12). Moreover, the emission efficiencies of the $S_1$ states for SM16 and EC9 show no significant disparities (0.9% *vs.* 0.4%, Supplementary Fig. 13). Therefore, in the D18/EC9/SM16 ternary blend, the values of $k_{CT}$, $k_{S_1}$, and $\eta_{S_1}$ are expected to remain relatively consistent despite variations in the SM16 content.

Then we constructed ternary solar cells based on D18/EC9/SM16, and anticipated that $k_{DS}$ decreases with increasing SM16 content, which then gives rise to an increase in $EQE_{EL}$. This is indeed the case as shown in Fig. 5c: $EQE_{EL}$ of the ternary solar cell increases from 0.02% to 0.04%, with the increase of SM16 content from 0% to 10%. Since the lifetime of charge carriers also increases with SM16 content, as confirmed by TPV measurements (Supplementary Fig. 14), the overall voltage loss is reduced and the $V_{OC}$ is increased for the ternary solar cells (Fig. 5c). It is noted that the degree of reduction in overall voltage loss exceeds that anticipated from the increase in $EQE_{EL}$, suggesting that there could be additional reasons for the increase in $V_{OC}$ with the addition of SM16. Nevertheless, neither the peak $EQE_{PV}$ (Fig. 5b) nor the *FF* (Table 2) of the ternary solar cell is significantly reduced by the

increased SM16 content, even with a high SM16 content of 20%. This is attributed to the similar molecular structures of SM16 and EC9, leading to similar morphological properties of the active layers, regardless of the SM16 content (Supplementary Fig. 15). As a result, the *PCE* of the solar cell significantly increases with the addition of SM16: The *PCE* increases from 17.4% to 19.2% as the SM16 content increases from 0% to 10% (Fig. 5d). The basic photovoltaic performance parameters of the solar cells with different SM16 contents are summarized in Table 2, and the *J-V* curves are provided in Supplementary Fig. 16.

Finally, referring to Eq. (5), we can observe that the dual-acceptor strategy possesses the potential to notably enhance $EQE_{EL}$, particularly when incorporating a secondary acceptor (A2) with an $S_1$ state energy close to the $E_{CT}$ of the D/A2 blend but substantially higher than the $S_1$ state energy of the other acceptor (A1). This expectation arises because using A2 in this manner elevates $E_{CT}$ and reduces both $k_{CT}$ and $k_{DS}$ of the ternary blend, thereby leading to a significant $EQE_{EL}$ increase. However, as elaborated in Supplementary Note 6, utilizing A2 with a higher $S_1$ state energy does not positively impact the performance of D/A1/A2 solar cells due to excited state energy transfer between

A2 and A1. Hence, we formulate the subsequent selection rule for A2 to modulate $k_{DS}$ and enhance solar cell performance: The $S_1$ state energy of A2 should be lower than that of A1, as depicted in case 2 of Fig. 2.

## Discussion

The above results affirm the general effectiveness of the dual-acceptor strategy in reducing $k_{DS}$ within OSCs. Furthermore, the reduced $k_{DS}$ can indeed result in an increase in $EQE_{EL}$, a decrease in $V_{NR}$, and an improvement of $PCE$ for high-efficiency NFA OSCs, provided that the electronic properties of the active materials align with the characteristics depicted in case 2 of Fig. 2. It is crucial to emphasize that the preference for the dual-acceptor approach over dual-donor stems from the significantly lower $S_1$ state energy of the acceptor in state-of-the-art OSCs compared to that of the donor. Consequently, the excited state concentration of the donor is negligible. For OSCs where the $S_1$ state energy of the acceptor surpasses that of the donor, as seen in fullerene solar cells, a similar dual-donor strategy is projected to effectively decrease the $k_{DS}$ of the blend system, enhance $EQE_{EL}$, and improve overall solar cell performance. Furthermore, Eq. (5) prompts us to acknowledge that the enhancement of $EQE_{EL}$ achieved by reducing $k_{DS}$ is highly reliant on the emission efficiency of the $S_1$ state. Currently, the emission quantum efficiency of the most emissive acceptors in thin films for OSCs remains below 1%. Hence, in order to fully harness the benefits of the dual-acceptor strategy, there is a pressing need to develop acceptor materials with higher emission efficiency.

To conclude, this study undertook a reevaluation of the dynamic processes governing excited states in OSCs, incorporating the repopulation and redissociation of the $S_1$ state of the acceptor. The findings reveal that the dissociation rate constant of the $S_1$ state, denoted as $k_{DS}$, traditionally associated with the generation efficiency of free charge carriers, plays a pivotal role in determining the non-radiative voltage loss within OSCs. Specifically, our results illustrate that instead of pursuing maximal $k_{DS}$ values for achieving high-efficiency OSCs, a deliberate reduction in $k_{DS}$ can lead to reduced non-radiative voltage loss and an enhanced overall device performance. In this regard, we subsequently introduced the dual-acceptor strategy, which centers on modifying the dynamic properties of excited states to effectuate a precisely controlled reduction in $k_{DS}$ within real-world OSCs. We successfully demonstrated that, for high-efficiency binary OSCs, incorporating a secondary acceptor with an $S_1$ state energy closely aligned to the $E_{CT}$ of the blend formed by the donor and the secondary acceptor enables a decrease in $k_{DS}$. This subsequently results in reduced non-radiative voltage loss and an improved solar cell performance. Consequently, the theoretical insights and experimental approach outlined in this study carry significant implications for overcoming the voltage bottleneck in OSCs, and for generating innovative concepts pertaining to the synthesis of high-efficiency photovoltaic materials, as well as the engineering of high-performance organic donor-acceptor blends.

## Methods

### Materials and device fabrication

The active materials, PM6, PM7, D18, IT4F, Y5, Y5OD, BTA3, and EC9 were purchased from Solarmer materials Inc (China). PC$_{71}$BM was purchased from Solenne BV. SM16 were synthesized according to the literature (*Adv. Funct. Mater.* 32, 2107756 (2022)). PFN-Br used in this work was purchased from Sigma Aldrich. PEDOT:PSS 4083 was purchased from Heraeus. 1,8-diiodooctane (DIO), 1-chloronaphthalene (CN), chlorobenzene (CB), and chloroform (CF) were purchased from Sigma-Aldrich.

Organic solar cells with a standard architecture of ITO/PEDOT:PSS 4083/active layer/PFN-Br/Ag were fabricated in this work. The ITO substrate was cleaned with acetone, isopropanol, and ethanol solvent in sequence within an ultrasonic cleaner for a

duration of 15 min. Subsequently, the ITO surface underwent a 10-minute plasma treatment. Then, PEDOT:PSS 4083 (purchased from Heraeus) solution was spin-coated (WS-650Mz-23NPPB, Laurell) on the ITO substrate at a speed of 4000 rpm, followed by annealing on a hot plate (IKA RCT digital) at 150 °C for 20 min. The thickness of PEDOT:PSS 4083 layer was approximately 30 nm. The PM7/IT4F/Y5 and PM7/IT4F/Y5OD active solutions were prepared by dissolving the materials at 20 mg mL$^{-1}$ in chlorobenzene (CB) (purchased from Sigma-Aldrich) with different D/A1/A2 weight ratios. In addition, 0.5% DIO was added in the solution as the solvent additive. Subsequently, each solution was heated to 80 °C and stirred at a rate of 1000 rpm utilizing a magnetic stirrer hotplate for 8 h. The solution was allowed to stand for half an hour and then cooled to room temperature prior to use. Then, the solutions for the active layer were spin-coated (2200 rpm) on top of the PEDOT:PSS 4083 coated substrate, in a nitrogen-filled glove box (Mbraun), and the resulting active layers were annealed at 110 °C for 10 min on a hotplate. The thickness of the active layers was about 100 nm. The D18/SM16/EC9 solutions were prepared by dissolving the active materials at a concentration of 16 mg mL$^{-1}$ in chloroform (CF) (from Sigma-Aldrich) with different D/A1/A2 weight ratios. Additionally, 0.5% CN (from Sigma-Aldrich) was added in the solution as an additive. All solutions were heated at 60 °C and stirred at 800 rpm on a magnetic stirrer hotplate for 4 h. The active layer solutions were spin-coated on the PEDOT:PSS 4083 coated substrates at 3000 rpm in a glove box, then the active layers were annealed at 90 °C for 5 min on a hotplate. The thickness of the active layers was about 90 nm. The PM7/BTA3/IT4F solution was prepared using CB as the solvent and the solution concentration was 20 mg mL$^{-1}$, supplemented with 0.5% DIO as an additive. Prior to use, the solutions underwent heating at 80 °C and they were stirred at 1000 rpm on a magnetic stirrer hotplate for a duration of 8 h. The active layers were spin-coated onto the PEDOT:PSS 4083 coated substrates at 2500 rpm, and they were annealed at 110 °C for 10 min on a hotplate. The thickness of the active layers was about 100 nm. Following the deposition of the active layers, the PFN-Br solution (0.5 mg mL$^{-1}$ in methanol, purchased from Sigma-Aldrich) was spin-coated onto the active layers at a speed of 3000 rpm. Then, a 110 nm thick Ag electrode was evaporated onto the active layers in a thermal evaporator (Mbraun) under a vacuum pressure of $3 \times 10^{-6}$ mbar. The active area of the devices determined by an optical microscope was 0.04 cm$^2$. Finally, the solar cells were encapsulated by employing glass slides in conjunction with UV-curing adhesive.

### Device characterization

***J-V* characterization.** Photovoltaic performance parameters of the organic solar cells studied in this work were characterized under simulated AM1.5 solar illumination (100 mW cm$^{-2}$) from a Newport Oriel VeraSol-2™ Class AAA solar simulator. Prior to testing, the illumination intensity was calibrated by a standard silicon solar cell and a set of low pass filters. The *J-V* curves were measured by a Keithley 2400 source meter. The $J_{SC}$ values from *J-V* characterizations and those derived from the $EQE_{PV}$ spectra exhibited a marginal discrepancy (<5%).

**Highly sensitive $EQE_{PV}$ measurements.** The $EQE_{PV}$ spectra were measured using a halogen lamp (LSH-75, 250 W, Newport), a monochromator (CS260-RG-3-MC-A, Newport), equipped with a series of long-pass filters (600 nm, 900 nm, 1000 nm), a front-end current amplifier (SR570, Stanford), a phase-locked amplifier (SR830, Stanford Instrument), and an optical chopper (173 Hz). Calibration of the light intensity was performed using a standard Si detector (Hamamatsu s1337-1010BR). A background illumination of 100 mW cm$^{-2}$ was applied during the measurements. The illuminated area was approximately 0.4 mm$^2$, achieved through the use of a focus lens and an optical aperture.

***EQE<sub>EL</sub> measurements.*** *EQE<sub>EL</sub>* measurements were conducted through a custom-built setup, which consists of a Keithley 2400 to apply electric current to the device in a dark environment. Photons emitted from the device was recorded using a Si detector (Hamamatsu s1337-1010BR) in conjunction with a Keithley 6482 Picoammeter.

**Photoluminescence measurements.** A Super-continuous white laser (SuperK EXU-6, NKT photonics) coupled to a narrowband filter (LLTF Contrast SR-VIS-HP8, LLTF Contrast SR-SWIR-HP8, NKT photonics) was used to excite organic thin films, and emission from the films was collected by an optical fiber (BFL200LS02, Thorlab), and directed to a fluorescence spectrometer (KYMERA-328I-B2, Andor Technology). Fluorescence signals were captured by either a Si EMCCD camera (DU970P-BVF, Andor Technology) covering the range of 400–1000 nm or an InGaAs EMCCD camera (DU491A-1.7, Andor Technology) covering the range of 1000–1500 nm. Optical losses in the fibers and the spectrometer were corrected using a standard light source (HL-3P-CAL, Ocean Optics Germany GmbH).

**Electroluminescence measurements.** Electroluminescence spectra were captured using a fluorescence spectrometer (KYMERA-328I-B2, Andor Technology LTD), a Si EMCCD camera (DU970P-BVF, Andor Technology), and an InGaAs EMCCD camera (DU491A-1.7, Andor Technology). Injection current was supplied to the sample device through a direct-current meter (PWS2326, Tektronix) in dark.

**Transient absorption spectroscopy.** The fs-TA spectra were measured by using a Helios-pump-probe setup (Ultrafast Systems), with an amplified 1030 nm fs laser (Pharos, Light Conversion, 200 fs, 200 μJ) with an effective repetition rate of 1 kHz. The probe white light was generated via non-linear phenomenon with a photonic crystal, yielding broadband probe light across the visible to NIR region from 400 to 950 nm. The pump pulse was generated with an optical parametric amplifier (Orpheus-F, Light Conversion), while the excitation intensity was tuned by density filters. The measurements were done using an excitation fluence of $1.7 \, \mu J \, cm^{-2}$, Samples were spin-coated on glass substrate and measured at the ambient condition without room light.

**Absorption measurements.** The absorption spectra of organic thin films were recorded using a UV-Vis spectrometer (Lambda 950, PerkinElmer).

**Thicknesses measurements.** The thicknesses of the organic thin films were measured utilizing a surface profilometer (KLA-Tencor P-7 Stylus Profileror).

**Atomic force microscope.** AFM images of the active layers were recorded by MFP-3D-BIO™, Asylum Research, in a tapping-Mode.

**Transmission electron microscope.** TEM images were taken using a JEM 2100 transmission electron microscope with an accelerating voltage of 200 kV.

### Reporting summary
Further information on research design is available in the Nature Portfolio Reporting Summary linked to this article.

## Data availability
All data generated in this study are provided within the article and the supplementary information. Source data are provided with this paper.

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

## Acknowledgements

Z.M., Z.B. and Z.T. acknowledges funding from the National Natural Science Foundation of China (22375035, 51973031, 51933001, 52073056). Z.T. acknowledges funding from the Natural Science Foundation of Shanghai (22ZR1401900). Z.M. and Z.T. acknowledges funding from the Fundamental Research Funds for the Central Universities (2232022A13, 2232021A09). H.W. acknowledges funding from the Graduate Student Innovation Fund of Donghua University (CUSF-DH-

D-2023006). W.T. acknowledges funding from the European Union's Horizon 2020 research and innovation program under grant agreement no. 851676 (ERC StGrt).

## Author contributions

The project was designed by Z.T. and Z.M., and supervised by Z.T., K.V., W.T, Z.B., and Z.M.; H.W. and H.L. fabricated and optimized the OPV devices, and performed *J-V, EQE$_{PV}$, EQE$_{EL}$*, UV-vis, steady state PL, TPV, and EL measurements, under the supervision of Z.T., Z.M., and Z.B.; H.W., X.Z., and G.Z. prepared thin films samples and performed fs-TA measurements and analyses, under the supervision of Y.L. and F.L.; H.P. and H.Y.W. performed AFM and TEM measurements under the supervision X.F.; H.W., Z.T., K.V., and W.T. derived the analytic equations for *EQE$_{EL}$* and *IQE*; Z.T. and Z.M. wrote the manuscript, and all authors contributed to the discussion and the finalizing of the manuscript.

## Competing interests

The authors declare no competing interests.
