## [Peer Review File · Nature Communications]

Decreasing Exciton Dissociation Rates for Reduced Voltage Losses in Organic Solar CellsREVIEWER COMMENTS

Reviewer #1 (Remarks to the Author):

Addressing the low open circuit voltages in organic solar cells relative to their optical band gap is now the primary concern for increasing the power conversion efficiencies of these systems to 20% and beyond. However, progress on addressing this issue in the field has arguably stalled, with PCEs stagnating at around ~19% for the last 2 years. Therefore, it is necessary to develop new strategies to further improve Voc beyond simply reducing the energetic offset between the singlet exciton and charge transfer states (which, as the authors demonstrate, eventually compromises the charge generation efficiency) and trying to improve the luminescence efficiency of the low gap component (a formidable task for low band gap organic emitters due to the energy gap law).

In their work, the authors propose a new tactic to improve Voc, involving the addition of a small fraction of a second acceptor material into the binary blend. What makes this distinct from a more traditional ternary strategy is that the first acceptor should have a larger energetic offset with the donor, whilst the second acceptor should have a smaller offset, and importantly a S1 energy very close to the CT state between the other components. When this is achieved in a model PM7:IT4F:Y5 blend (and the closely related PM7:IT4F:Y5OD system), the authors observe a small but significant increase in the EQE_EL and thus Voc without sacrificing the Jsc or FF. In general, I find the strategy proposed by the authors to be novel and potentially of high importance to the field. However, though the second system presented (D18:EC19:SM16) does indeed offer excellent performance, I don't think that the gains can be as clearly ascribed to the authors' strategy as in the PM7:IT4F:Y5 system. Furthermore, there are significant issues with the transient absorption spectroscopy data, specifically related to the extremely high signal sizes measured and the spectral deconvolution strategy employed. Nonetheless, if these issues can be addressed, I think the paper could provide an important contribution to the field and thus be suitable for publication in Nature Communications. Below, I have provided some comments that I hope will assist the authors with revising their manuscript.

1. In the introduction, the authors state: '...has resulted in power conversion efficiencies (PCE) of organic solar cells (OSCs) based on the bulk-heterojunction (BHJ) concept approaching and exceeding 20%.'

However, PCEs over 20% have only been achieved for tandem devices. The authors should make this caveat clear to avoid misleading the reader who may believe that this means PCEs >20% have been achieved in the much more common single junction devices.

2. In line 83, page 4, the name 'Gillett' is misspelled.

3. When deriving equation 3 for the EQE_EL of an OSC, the authors state: 'In NFA OSCs, electroluminescence (EL) also includes S1A emission, resulting from the back transfer of CT state to the singlet state. This introduces S1A as an additional decay channel for free charge carrier recombination in NFA OSCs.'

However, this additional pathway for CT states is only part of the story. Despite noting earlier that recombination to the molecular triplet exciton (T1) states can be a significant recombination pathway for CT states, this has been neglected in their subsequent analysis. In the simple spin statistical approximation, 75% of free charge recombination events can form the 3CT state that acts as the intermediate to T1 formation, and the actual T1 formation yield can exceed this (see: 10.1038/s41586-021-03840-5, 10.1021/acsnano.6b06211, 10.1002/aenm.202301357). Thus, it is not possible to derive an accurate expression for EQE_EL that doesn't include the formation of molecular triplet excitons from CT states.

Nevertheless, I appreciate the essence of what the authors have tried to do and still think there is value in the equations they have defined when considering specifically recombination via the spin singlet manifold, which of course must be simultaneously optimised as T1 recombination is suppressed. Therefore, if the authors wish to retain the subsequent derivations based on equation 3, perhaps they could consider including a simple pre-factor related to the T1 recombination fraction. For example $(1-\alpha)$, where α is the fraction of CT recombination events that form T1 states and thus do not result in photon emission.

4. The measured signal sizes in the TA ($\Delta T/T \sim 3 \times 10^{-2}$ in PM7:Y5 and $\sim 1 \times 10^{-2}$ PM7:IT4F) are far too large for any meaningful information to be extracted. At such a large signal size, the excited state population will be so high that there will be significant excess recombination even before the charge transfer process has completed. This becomes very problematic when the authors then attempt to extract the S1 to CT conversion timescales (k_{DS}) from the TA data, as the excessively high excitation fluence will lead to an underestimation of this parameter. To address this, the authors should reduce their excitation fluence to find the linear regime where there is no excess non-geminate recombination over the timescales of their TA experiment ($\sim 1-2$ ns). This can be achieved by performing a detailed excitation fluence series and finding the point at which the two lowest fluences show identical TA kinetics. Only then, can an accurate value for the S1 to CT conversion be obtained. From my experience, I imagine that this will be in the regime of a $\Delta T/T \sim 5 \times 10^{-4}$, which is 2 orders of magnitude smaller than the data presented.

5. The TA excitation fluences have not been provided in the current paper. They should be included in the revised version.

6. To determine the S1 to CT conversion timescales, the authors use an MCR analysis to extract the different spectral components. Whilst I understand that this and similar spectral deconvolution techniques are used fairly regularly for analysing TA data, I am always a little concerned that they are treated as a 'black box' and the results taken as correct without a deeper critical analysis of the deconvolution effectiveness. Therefore, I would ask that the authors provide some additional error analysis for their deconvolution; for example, a 2D plot of the residual of the extraction as a function of both wavelength and time. This would allow the reader to better examine the accuracy of their deconvolution.

7. The authors show the contributions from different spectral components in their deconvolution analysis in SI Figure 4. It would first be helpful if these were plotted on a log timescale, as the early time spectral evolutions basically appear as a straight line on the hundreds of picoseconds scale linear time axis. But, beyond this, I would ask how the authors can physically justify using (by my count) 10 contributions to reconstruct the observed TA spectrum? Then, presumably choosing to use only the one which they (arbitrarily) determine corresponds to the charge transfer timescale for further analysis. The effect of including multiple weaker contributions is effectively to decrease the residual though the addition of multiple unphysical spectral components to account for any discrepancy between the dominant fit components and the actual TA data. It would be useful for the authors to show each of the 10(?) spectral contributions so the reader can further assess the validity of their spectral deconvolution.

8. One might expect that the fit to 2 species (pre- and post- charge transfer) should be largely sufficient to describe the observed TA data. This can be seen by the fact that two species provide dominant contributions to the TA spectrum in SI Figure 4. What happens if the authors instead limit their deconvolution to 2 species as opposed to 10? Does the error in the fit significantly increase then? And what does a 2D plot of the residual of the extraction as a function of both wavelength and time look like now?

9. To avoid any questions about the deconvolution altogether, the authors could also instead just use timescales taken from the raw TA data kinetics, which by definition will always be more trustworthy than any sort of spectral deconvolution. For example, PBDB-T family donor polymers have a fairly broad hole polaron PIA that extends beyond the spectral features of the NFA in the TA on the blend. An example of this can be seen in Figure 3b in this paper: [10.1039/D1EE03565G](https://doi.org/10.1039/D1EE03565G). Here, a kinetic taken from ~ 1020 nm would yield a kinetic that accurately tracks the rise of the donor polymer hole polaron PIA (i.e. the conversion of S1 into CT states/charges) without any significant overlap with the other spectral features in the blend. Thus, a similar analysis in the authors' work could provide an alternative way to determine the S1 to CT conversion timescale without having to rely on spectral deconvolution and the associated issues.

10. To examine the CT state generation efficiency of different acceptors, the authors then compare the S1 to CT conversion timescales from the TA (which are themselves already flawed due to the excessively

high signal; see above) with the measured PL lifetime by TCSPC. Whilst I understand the logic behind this, unfortunately the complicated behaviour of the Y-series NFAs means this is not a valid strategy. Specifically, the measured excited state lifetime of Y-series NFA by TA (~300 ps) is typically much shorter than the measured PL lifetimes (~1 ns). Whilst the reason for this discrepancy is unclear (and likely deserving of a paper by itself), I would recommend that the authors measure the TA of the neat acceptor films (again with a fluence series to determine the excited state lifetime in the linear regime), and then use this TA lifetime instead of the PL lifetime in their CT generation efficiency analysis.

11. It would be good to see some TA measurements performed on the three-component systems with different wt% of the second acceptor to confirm what the authors propose. Specifically, that the addition of a small amount of the lower offset NFA (for example, Y5 in the PM7:IT4F system) does indeed result in a small increase in the S1 to CT conversion timescale that at first has minimal impact on the efficiency of this process, but eventually results in a reduction of the charge generation yield. As the TA spectrum will become more complicated upon the addition of the second acceptor, perhaps it might be necessary to track the hole polaron PIA, as discussed above, instead of relying on the spectral deconvolution.

12. In addition, I would be curious to learn more about the dynamics of the two electron acceptor materials in the three-component blend. For example, if the film was excited at something like 580 nm to preferentially pump the PM7 donor, one might expect the initial electron transfer step to largely proceed from PM7 to IT4F, due to the excess wt% of this component vs Y5 in the highest performing system. However, do the authors see any evidence for a delayed increase in the population of the Y5 component (either the GSB or the S1 PIA), potentially indicating some sort of equilibration forming between the PM7:IT4F CT state and the Y5 S1, as suggested by the authors?

13. Whilst the device performance changes in PM7:IT4F:Y5 are quite convincing, I think the situation isn't as clear cut in D18:EC9:SM16. For example, there is only a small reduction in the V_{nr} (0.12 V) from 0.210 to 0.198 V with 10 wt% of SM16, yet the V_{oc} increase is much larger (0.37 V) from 0.869 to 0.906 V. In addition, there is also a noticeable improvement in the FF from 74.9% to 78.5%. Both of these factors, which seem to be separate from the mechanism involving a reduced V_{nr} proposed by the authors, contribute much more significantly to the improved PCE from 17.4% to 19.2%. Thus, I would caution the authors about ascribing the performance improvement seen here, as they could instead be attributed to morphological effects.

14. Towards the end of the manuscript, the authors also propose that a similar third acceptor strategy should also work in fullerene systems. Whilst I accept that fullerene acceptor OSCs are no longer state of the art, I think if the authors could also demonstrate that their strategy could indeed work in a large offset fullerene blend, it would be a very convincing model system. Thus, I would ask if the authors could try to replicate their strategy in a fullerene system as further evidence for its effectiveness. This would

also go some way to allaying my concerns about drawing too strong conclusions from the D18:EC9:SM16 system, which I find to be a less convincing example of their proposed strategy.

Reviewer #2 (Remarks to the Author):

The paper demonstrates that the device electroluminescence quantum efficiency, a critical factor in mitigating non-radiative voltage losses and further improving the performance of organic solar cells (OSCs), is influenced by the dissociation rate constant of singlet states. This is not the only parameter in mitigating non-radiative voltage losses, so that controlling such parameter will not warrant a substantial rise in efficiency for OSCs. The paper indicates an experimental design which would push the limit, but does not clearly quantify its impact. The paper is well written, deserves to be published, but certainly lacks the urgency and broad impact needed for a Nature communications paper. I suggest to reject it.

Reviewer #3 (Remarks to the Author):

The authors present a hypothesis for OPV solar cells, which suggests that slowing down the charge transfer rate (from S1 to CT state) to a certain extent can be beneficial for power conversion efficiency, as it enhances EQE_EL without compromising current generation. They base this hypothesis on a simulation of the PCE as a function of the rates of different CT and back transfer processes. They experimentally support their hypothesis by testing three ternary OPV blends containing two acceptors, which they claim reduces the CT rate. The story could in principle be of interest, even for a high impact journal such as Nat. Commun., but it cannot be accepted currently. Especially since the superior performance of ternary blends is not a new concept, they need to prove their hypothesis much more strongly. Currently, there are serious problems with the data, the interpretation and the substantiation of the conclusions.

1) The assumptions made when deriving their hypothesis are too simplistic. The authors say that a low CT-S1 offset leads to slow charge transfer rates. This is not necessarily true when the re-organization energy is low. Many studies show ultrafast charge transfer at very low offsets. Also, they authors claim that the back transfer rate is higher when the forward rate is slower. Again, this is not necessarily true. If the Marcus curve peaks at zero driving force, both rates will be maximized, leading to an equilibrium between the two states. The authors should revise their understanding of Marcus theory.

2) The TA measurements cannot be taken seriously, because the fluence used was way too high. To avoid non-linear artefacts in NFA blends, TA signals need to be below $1 \cdot 10^{-3}$ OD. Here, they are about 20 times higher.

3) The authors claim a S1 decay rate of 800 ps for one of the acceptors, justifying that slower CT rates are not a problem. However, they find this time constant based on nanosecond-resolved PL measurements. They miss out all the fast components of the exciton decay. Typically, the S1 lifetime in films is very multi-exponential with fast components that can very well compete with the CT time.

4) The structural characterization by AFM (and by using molecules of similar structure) is not at all sufficient. This can have very profound influence on the photophysical processes. For example, the charge transfer rates of 20 ps and 70 ps are quite slow (usually it is on the <500 fs scale for efficient systems). This is likely because there is exciton diffusion through neat donor and acceptor domains before the CT step. The complex morphology of the ternary blends must be understood and controlled in detail in order to exclude this.

5) It is not new that ternary blends perform better and yield high efficiencies. The authors present a new hypothesis for the reason. However, they do not rule out other possible reasons for the improved performance. In particular, the morphology could have an important effect. Also, the charge generation is likely complicated by energy transfer processes between the different blend constituents, and there might be charge hopping between the two acceptors. This needs to be investigated to make sure the hypothesis of the authors is correct.

REVIEWER COMMENTS

Reviewer #1 (Remarks to the Author):

Addressing the low open circuit voltages in organic solar cells relative to their optical band gap is now the primary concern for increasing the power conversion efficiencies of these systems to 20% and beyond. However, progress on addressing this issue in the field has arguably stalled, with PCEs stagnating at around ~19% for the last 2 years. Therefore, it is necessary to develop new strategies to further improve Voc beyond simply reducing the energetic offset between the singlet exciton and charge transfer states (which, as the authors demonstrate, eventually compromises the charge generation efficiency) and trying to improve the luminescence efficiency of the low gap component (a formidable task for low band gap organic emitters due to the energy gap law).

In their work, the authors propose a new tactic to improve Voc, involving the addition of a small fraction of a second acceptor material into the binary blend. What makes this distinct from a more traditional ternary strategy is that the first acceptor should have a larger energetic offset with the donor, whilst the second acceptor should have a smaller offset, and importantly a S1 energy very close to the CT state between the other components. When this is achieved in a model PM7:IT4F:Y5 blend (and the closely related PM7:IT4F:Y5OD system), the authors observe a small but significant increase in the EQE_{EL} and thus Voc without sacrificing the Jsc or FF. In general, I find the strategy proposed by the authors to be novel and potentially of high importance to the field. However, though the second system presented (D18:EC19:SM16) does indeed offer excellent performance, I don't think that the gains can be as clearly ascribed to the authors' strategy as in the PM7:IT4F:Y5 system.

Furthermore, there are significant issues with the transient absorption spectroscopy data, specifically related to the extremely high signal sizes measured and the spectral deconvolution strategy employed. Nonetheless, if these issues can be addressed, I think the paper could provide an important contribution to the field and thus be suitable for publication in Nature Communications. Below, I have provided some comments that I hope will assist the authors with revising their manuscript.

Our response: We highly appreciate the reviewer for the overall positive comments and provision of an expert view on decay of excited states via triplet states and our TA measurements. For the revised manuscript, we have derived the expression for EQE_{EL} in the case of having decay via triplet states, and redone the TA measurements following the protocol suggested by the reviewer, and below, we provide our detailed response to the reviewer's comments.

1. In the introduction, the authors state: '...has resulted in power conversion efficiencies (PCE) of organic solar cells (OSCs) based on the bulk-heterojunction (BHJ) concept approaching and exceeding 20%. However, PCEs over 20% have only been achieved for tandem devices. The authors should make this caveat clear to avoid misleading the reader who may believe that this means PCEs >20% have been achieved in the much more common single junction devices.

Our response: We have revised our statement regarding the record PCE achieved for organic solar cells as follows

“.....has resulted in power conversion efficiencies (PCE) of organic solar cells (OSCs) based on the bulk-heterojunction (BHJ) concept approaching and exceeding 19%”.

2. In line 83, page 4, the name ‘Gillett’ is misspelled.

Our response: We have corrected the spelling of the name.

3. When deriving equation 3 for the EQE_{EL} of an OSC, the authors state: ‘In NFA OSCs, electroluminescence (EL) also includes $S1^A$ emission, resulting from the back transfer of CT state to the singlet state. This introduces $S1^A$ as an additional decay channel for free charge carrier recombination in NFA OSCs.’

However, this additional pathway for CT states is only part of the story. Despite noting earlier that recombination to the molecular triplet exciton (T1) states can be a significant recombination pathway for CT states, this has been neglected in their subsequent analysis. In the simple spin statistical approximation, 75% of free charge recombination events can form the 3CT state that acts as the intermediate to T1 formation, and the actual T1 formation yield can exceed this (see: 10.1038/s41586-021-03840-5, 10.1021/acsnano.6b06211, 10.1002/aenm.202301357). Thus, it is not possible to derive an accurate expression for EQE_{EL} that doesn’t include the formation of molecular triplet excitons from CT states.

Nevertheless, I appreciate the essence of what the authors have tried to do and still think there is value in the equations they have defined when considering specifically recombination via the spin singlet manifold, which of course must be simultaneously optimised as T1 recombination is suppressed. Therefore, if the authors wish to retain the subsequent derivations based on equation 3, perhaps they could consider including a simple pre-factor related to the T1 recombination fraction. For example $(1-\alpha)$, where α is the fraction of CT recombination events that form T1 states and thus do not result in photon emission.

Our response: We appreciate the reviewer for highlighting the decay of excited states via triplet states as a significant loss factor. In the Supplementary Note 1 of the revised manuscript and below, we provide the derivation of the expression for EQE_{EL} in the case of the excited states partially decaying via triplet state. We have also revised the main text of the manuscript.

“In case of having non-geminate decay of excited states via triplet states, for NFA OSCs, we can express the EQE_{EL} as

$$EQE_{EL} = \frac{n_{S1}k_r^{S1} + n_{CT1}k_r^{CT1}}{n_{S1}k_{S1} + n_{CT1}k_{CT1} + n_{CS}k_{CS}^{CT3}}$$

where n_{CT1} , n_{S1} , and n_{CS} are the equilibrium concentrations of singlet CT state, $S1^A$, and charge separated state, respectively, k_{CT1} and k_{S1} are the decay rate constants of singlet CT state and $S1^A$, respectively. k_{CS}^{CT3} is the rate constant for the transfer of charge separated state to the triplet CT state. Here, we assume that the excited states are directly lost upon forming the triplet CT states from charge-separated states. This

assumption is based on the expectation that the conversion of triplet CT states to the molecular triplet states (T1) is much faster than the conversion of triplet CT states to the singlet CT states or the dissociation of triplet CT states into free charge carriers (Nature 2021, 597, 666). A schematic illustration of the excited state dynamics for NFA OSCs under EL or EQE_{EL} measurements is presented in Supplementary Figure 17, and below in Figure R1.

Figure R1. Dynamics of excited states in NFA OSCs under EL or EQE_{EL} measurement in the case of having excited states partially decayed via triplet states. $S1^A$, CT1, CT3, and CS are abbreviations for acceptor singlet, singlet charge transfer, triplet charge transfer, and charge separated states, respectively.

Assuming a simple spin statistical approximation, 75% of free charge carriers recombine via CT3, we have

$$n_{CS}k_{CS}^{CT3} = 3n_{CS}k_{CS}^{CT1}$$

where k_{CS}^{CT1} is the rate constant for the transfer of charge separated state to the singlet CT state. Then, the expression for EQE_{EL} becomes,

$$EQE_{EL} = \frac{n_{S1}k_r^{S1} + n_{CT1}k_r^{CT1}}{n_{S1}k_{S1} + n_{CT1}k_{CT1} + 3n_{CS}k_{CS}^{CT1}}$$

Because the generation rate of CT1 must be in equilibrium with the disappearing rate of CT1, we derive

$$n_{CT1}(k_{BK} + k_{CT1}^{CS} + k_{CT1}) = n_{S1}k_{DS} + n_{CS}k_{CS}^{CT1}$$

where k_{BK} is the rate constant for the transfer of CT1 to $S1^A$, k_{CT1}^{CS} is the rate constant for the dissociation of CT1 into free charge carriers, k_{CT1} is the rate constant for the decay of CT1 to the ground state, and k_{DS} is the rate constant of the dissociation of $S1^A$. This gives

$$n_{CS}k_{CS}^{CT1} = n_{CT1}(k_{BK} + k_{CT1}^{CS} + k_{CT1}) - n_{S1}k_{DS}$$

Then, the expression for EQE_{EL} becomes

$$EQE_{EL} = \frac{n_{S1}k_r^{S1} + n_{CT1}k_r^{CT1}}{n_{S1}k_{S1} + n_{CT1}k_{CT1} + 3n_{CT1}(k_{BK} + k_{CT1}^{CS} + k_{CT1}) - 3n_{S1}k_{DS}}$$

$$= \frac{\left(\frac{n_{S1}}{n_{CT1}}\right)k_r^{S1} + k_r^{CT1}}{\left(\frac{n_{S1}}{n_{CT1}}\right)k_{S1} + k_{CT1} + 3(k_{BK} + k_{CT1}^{CS} + k_{CT1}) - 3\left(\frac{n_{S1}}{n_{CT1}}\right)k_{DS}}$$

Since the generation rate of SI^A must be in equilibrium with the disappearing rate of SI^A , we derive

$$n_{S1}(k_{S1} + k_{DS}) = n_{CT1}k_{BK}$$

Thus,

$$\frac{n_{S1}}{n_{CT1}} = \frac{k_{BK}}{k_{S1} + k_{DS}}$$

Combining the above two equations, we have

$$\begin{aligned} EQE_{EL} &= \frac{\left(\frac{k_{BK}}{k_{S1} + k_{DS}}\right)k_r^{S1} + k_r^{CT1}}{\left(\frac{k_{BK}}{k_{S1} + k_{DS}}\right)k_{S1} + k_{CT1} + 3(k_{BK} + k_{CT1}^{CS} + k_{CT1}) - 3\left(\frac{k_{BK}}{k_{S1} + k_{DS}}\right)k_{DS}} \\ &= \frac{k_{BK}k_r^{S1} + k_r^{CT1}(k_{S1} + k_{DS})}{k_{BK}k_{S1} + k_{CT1}(k_{S1} + k_{DS}) + 3(k_{BK} + k_{CT1}^{CS} + k_{CT1})(k_{S1} + k_{DS}) - 3k_{BK}k_{DS}} \\ &= \frac{k_{BK}k_r^{S1}}{k_{BK}k_{S1} + k_{CT1}(k_{S1} + k_{DS}) + 3(k_{BK} + k_{CT1}^{CS} + k_{CT1})(k_{S1} + k_{DS}) - 3k_{BK}k_{DS}} \\ &\quad + \frac{k_r^{CT1}(k_{S1} + k_{DS})}{k_{BK}k_{S1} + k_{CT1}(k_{S1} + k_{DS}) + 3(k_{BK} + k_{CT1}^{CS} + k_{CT1})(k_{S1} + k_{DS}) - 3k_{BK}k_{DS}} \\ &= \frac{k_r^{S1}}{k_{S1} + \frac{k_{CT1}(k_{S1} + k_{DS}) + 3(k_{BK} + k_{CT1}^{CS} + k_{CT1})(k_{S1} + k_{DS}) - 3k_{BK}k_{DS}}{k_{BK}}} \\ &\quad + \frac{k_r^{CT1}}{k_{CT1} + \frac{k_{BK}k_{S1} + 3(k_{BK} + k_{CT1}^{CS} + k_{CT1})(k_{S1} + k_{DS}) - 3k_{BK}k_{DS}}{(k_{S1} + k_{DS})}} \end{aligned}$$

Defining that

$$\eta_{S1} = \frac{k_r^{S1}}{k_{S1}} \quad \text{and} \quad \eta_{CT1} = \frac{k_r^{CT1}}{k_{CT1}}$$

we derive

$$\begin{aligned} EQE_{EL} &= \frac{\eta_{S1}}{\frac{k_{CT1}(k_{S1} + k_{DS}) + 3(k_{BK} + k_{CT1}^{CS} + k_{CT1})(k_{S1} + k_{DS}) - 3k_{BK}k_{DS}}{k_{S1}k_{BK}} + 1} \\ &\quad + \frac{\eta_{CT1}}{\frac{k_{BK}k_{S1} + 3(k_{BK} + k_{CT1}^{CS} + k_{CT1})(k_{S1} + k_{DS}) - 3k_{BK}k_{DS}}{k_{CT1}(k_{S1} + k_{DS})} + 1} \end{aligned}$$

$$\begin{aligned}
&= \frac{\eta_{S1}}{k_{CT1}k_{S1} + k_{CT1}k_{DS} + 3k_{BK}k_{S1} + 3k_{CT1}^{CS}k_{S1} + 3k_{CT1}k_{S1} + 3k_{BK}k_{DS} + 3k_{CT1}^{CS}k_{DS} + 3k_{CT1}k_{DS} - 3k_{BK}k_{DS} + 1} \\
&+ \frac{\eta_{CT1}}{k_{S1}k_{BK}} \\
&+ \frac{k_{BK}k_{S1} + 3k_{BK}k_{S1} + 3k_{CT1}^{CS}k_{S1} + 3k_{CT1}k_{S1} + 3k_{BK}k_{DS} + 3k_{CT1}^{CS}k_{DS} + 3k_{CT1}k_{DS} - 3k_{BK}k_{DS} + 1}{k_{CT1}(k_{S1} + k_{DS})} \\
&= \frac{\eta_{S1}}{\left(\frac{k_{DS} + 1}{k_{S1}}\right) \cdot \left(4 + 3\frac{k_{CT1}^{CS}}{k_{CT1}}\right) + 4} + \frac{\eta_{CT1}}{\left(\frac{k_{DS} + 1}{k_{S1}}\right) \cdot 4 + \left(4 + 3\frac{k_{CT1}^{CS}}{k_{CT1}}\right)}
\end{aligned}$$

Finally, defining that

$$r_{DS} = \frac{k_{DS}}{k_{S1}}, \quad r_{BK} = \frac{k_{BK}}{k_{CT1}}, \quad \text{and } P = 4 + 3\frac{k_{CT1}^{CS}}{k_{CT1}}$$

we have

$$EQE_{EL} = \frac{\eta_{S1}}{\left(\frac{k_{DS} + 1}{k_{S1}}\right) \cdot P + 4} + \frac{\eta_{CT1}}{\left(\frac{k_{DS} + 1}{k_{S1}}\right) \cdot 4 + P}$$

The expression derived in the case of the excited state partially decaying via triplet states is similar to that derived assuming no decay of excited states via triplet states. The main difference lies in the addition of a positive parameter P (>4), in the denominators of the expression. This term is determined by the ratio between the rate constant for the dissociation of singlet CT state into free charge carriers (k_{CT1}^{CS}) and the rate constant for the direct decay of singlet CT state (k_{CT1}).

Then, we ran computer simulations using the expression for EQE_{EL} mentioned above and equation 7 in the main text of the manuscript to illustrate the impact of r_{DS} and r_{BK} on the performance of solar cells in the case of the excited state partially decaying via triplet states. As depicted in Supplementary Figure 18 and below in Figure R2, both EQE_{EL} and PCE are lower in the solar cell with excited states decaying via triplet states compared to that with no decay via triplet states, even when the P value is at its minimum of 4. However, regardless of the P value, we observed an increase in EQE_{EL} and PCE with the reduction of k_{DS} . Therefore, the conclusion that EQE_{EL} increases with the reduction of k_{DS} remains valid, in both cases with or without excited states decaying via triplet states.”

In the section 2.1 of the revised manuscript, we have added the clarification, “Note that when the decay of excited states via molecular triplet states is involved, equation 3 requires modification, resulting in a new expression for the relationship between EQE_{EL} and the rate constants, which differs slightly from equation 5. Nevertheless, the conclusion that EQE_{EL} increases with the reduction of k_{DS} remains valid. Further details on the derivation of equation 5, as well as the derivation of the expression with the decay via triplets included, can be found in Supplementary Note 1.”

Figure R2. EQE_{EL} and PCE as a function of r_{DS} and r_{BK} , **a)** calculated using equation 5 and equation 7 in the main text, and calculated using the expression for EQE_{EL} derived above and equation 7 in the main text, by assuming **b)** $P=4$, **c)** $P=40$, and **d)** $P=400$. η_{S1} and η_{CT1} of the solar cell are assumed to be 1% and $10^{-5}\%$, respectively. The radiative limit for V_{OC} , the upper limit for J_{SC} , and the FF of the solar cell are assumed to be 1.10 V, 32 mA cm $^{-2}$, and 80%, respectively.

4. The measured signal sizes in the TA ($\Delta T/T \sim 3 \times 10^{-2}$ in PM7:Y5 and $\sim 1 \times 10^{-2}$ in PM7:IT4F) are far too large for any meaningful information to be extracted. At such a large signal size, the excited state population will be so high that there will be significant excess recombination even before the charge transfer process has competed. This becomes very problematic when the authors then attempt to extract the S1 to CT conversion timescales (k_{DS}) from the TA data, as the excessively high excitation fluence will lead to an underestimation of this parameter. To address this, the authors should reduce their excitation fluence to find the linear regime where there is no excess non-geminate recombination over the timescales of their TA experiment ($\sim 1-2$ ns). This can be achieved by performing a detailed excitation fluence series and finding the point at which the two lowest fluences show identical TA kinetics. Only then, can an accurate value for the S1 to CT conversion be obtained. From my experience, I imagine that this will be in the regime of a $\Delta T/T \sim 5 \times 10^{-4}$, which is 2 orders of magnitude smaller than the data presented.

Our response: We appreciate the reviewer for noting that the fluence used for the TA measurement was too high. In the revised manuscript, we have repeated the TA measurements using different fluences, as shown below in Figure R3, and selected a fluence of $4.5 \mu\text{W}$ ($0.13 \mu\text{J cm}^{-2}$) for the subsequent measurements.

Figure R3. Time evolution of the PM7 ground state bleaching signals at 630 nm, extracted from the fluence dependent TA spectra of the PM7/IT4F blend, measured with a pump wavelength of 750 nm.

5. The TA excitation fluences have not been provided in the current paper. They should be included in the revised version.

Our response: The excitation fluence ($0.13 \mu\text{J cm}^{-2}$) used for the revised manuscript is now provided in the experimental section of the manuscript.

6. To determine the S1 to CT conversion timescales, the authors use an MCR analysis to extract the different spectral components. Whilst I understand that this and similar spectral deconvolution techniques are used fairly regularly for analysing TA data, I am always a little concerned that they are treated as a ‘black box’ and the results taken as correct without a deeper critical analysis of the deconvolution effectiveness. Therefore, I would ask that the authors provide some additional error analysis for their deconvolution; for example, a 2D plot of the residual of the extraction as a function of both wavelength and time. This would allow the reader to better examine the accuracy of their deconvolution.

Our response: In the revised manuscript, we followed the TA analysis protocol outlined in reference (Energy Environ. Sci., 2022,15, 1545), as suggested in comment 9 from the reviewer. Consequently, the discussion on MCR has been omitted in the manuscript. Nevertheless, we provide our response regarding the MCR analysis below.

Indeed, the MCR algorithm is a widely used method for TA data analysis. Mathematically, it is a matrix decomposition method used to derive the associated spectra and concentration evolution of different components in the experiment TA spectrum, yielding the residual E (or in other words, the lack of fit)

$$D = S_1(\lambda)C_1(t)^T + S_2(\lambda)C_2(t)^T + S_3(\lambda)C_3(t)^T + \dots + E \quad (\text{eq. R1})$$

In the above equation, D is the raw experimental data matrix, S represents the decomposed spectra for species involved in the photophysical process, and C denotes the concentration matrix for each component as a function of delay time (t) between the probe and pump light. The lack of fit (lof) function, calculated from the fitting residual matrix E , can be defined as follows:

$$lof (\%) = 100 \sqrt{\frac{\sum_{i,j} E_{i,j}^2}{\sum_{i,j} D_{i,j}}} \quad (\text{eq. R2})$$

where i and j are elemental positions in the matrix D and E .

It is crucial to estimate the number of components involved in TA data before conducting MCR analysis. Experimental techniques, such as time-resolved photoluminescence spectroscopy, can be employed to estimate the species involved after photon excitation. Additionally, since TA data is inherently a 2D matrix, principal component analysis using singular value decomposition (SVD) can mathematically be used to evaluate the number of components needed to represent the TA data matrix. Furthermore, a gradual SVD as a function of the delay time in the forward direction, also known as forward evolving factor analysis (EFA), can be used for obtaining the information about the starting time for each component contributing to the TA signal. By combining physical understanding, additional independent experimental results, and principal component analysis, one can estimate the number of components for the subsequent MCR analysis of a specific TA data matrix.

As depicted below in Figure R4 for the PM7/IT4F system, measured with an excitation wavelength of 750 nm to selectively excite the IT4F acceptor in the blend, one can clearly observe the ground state bleaching (GSB) of both IT4F (750 nm) and PM7 (550-650 nm). Additionally, a positive ΔA signals at 700 nm and 800-900 nm can be resolved, which is attributed to absorption from newly generated species after photoexcitation, such as the singlet excited state of IT4F and polaron absorption of PM7. The principal component analysis and EFA and the method to reduce the rotational ambiguity will be discussed in our response to comment 7 from the reviewer.

Here, to assess the error distribution after the MCR decomposition of the raw experimental data, we performed MCR analysis with two components on the raw experimental TA shown in Figure R4a for the PM7/IT4F system. The results are presented in Figure R4b. During the MCR alternative least square iterations, we observed that the factor lof , defined by eq. S2, quickly converges to its minimum after

several iterations (Figure R4d). Examining the 2D residual distribution shown in Figure R4c, we observed that the error is evenly distributed across the entire time scale and wavelength region.

Figure R4. MCR analysis for TA data of PM7/IT4F blend, excited at 750 nm with a fluence of $0.66 \mu\text{J}/\text{cm}^2$. **a)** Raw experimental data; **b)** Fit results from MCR analysis; **c)** Fitting residual; **d)** Lack of fit for different iterations.

7. The authors show the contributions from different spectral components in their deconvolution analysis in SI Figure 4. It would first be helpful if these were plotted on a log timescale, as the early time spectral evolutions basically appear as a straight line on the hundreds of picoseconds scale linear time axis. But, beyond this, I would ask how the authors can physically justify using (by my count) 10 contributions to reconstruct the observed TA spectrum? Then, presumably choosing to use only the one which they (arbitrarily) determine corresponds to the charge transfer timescale for further analysis. The effect of including multiple weaker contributions is effectively to decrease the residual though the addition of multiple unphysical spectral components to account for any discrepancy between the dominant fit components and the actual TA data. It would be useful for the authors to show each of the 10(?) spectral contributions so the reader can further assess the validity of their spectral deconvolution.

Our response: As discussed in our response to comment 6 from the reviewer, it is crucial to assess the number of components required to represent the experimental data before conducting the MCR analysis. Therefore, a singular value decomposition (SVD) is employed for the gradually increasing size of the TA data matrix with an increased delay time. This approach, also known as forward evolving factor analysis (EFA), helps estimate the necessary number of components for decomposing the experimental TA

data matrix. Once again, we illustrate this process using the TA data of the PM7/IT4F blend as an example.

In the forward EFA evaluation, various singular values are derived, with the 10 major singular values presented in below in Figure R5. Importantly, the first two singular values are at least one order of magnitude higher than the rest, indicating the presence of two major components contributing to the experimental TA spectra. Considering our photophysical understanding of the PM7/IT4F system, we anticipate that there are indeed two major components originating from the acceptor IT4F and the donor PM7. Consequently, we chose to utilize only 2 components, rather than all 10, for the MCR analysis and proceeded to decompose the experimental TA data. It is important to clarify that, although there are many singular values from the forward EFA analysis, only 2 components are used for the MCR analysis, as the reviewer may have misunderstood.

Figure R5. Forward EFA for the TA spectra of the PM7/IT4F blend shown in Figure R4a. Only ten major singular values are presented.

Below in Figure R6, the TA spectra determined from the MCR analysis are compared with the experimental raw data. We note that the MCR fit results obtained by using 2 components nicely represent the experimental raw data. We also stress that, for the MCR analysis, it is not a requirement to use only two components to decompose the TA data. One can use more than 2 components to decompose the experimental data for a lower *lof*. However, one must consider the physical meaning of each of these components to justify their addition, especially in cases where two components, easily assigned to well-understood physical processes, are already sufficient to decompose the experimental data with a sufficiently low error.

Figure R6. Comparison between the MCR fitting results (lines) and raw experimental data (symbols). The TA data of the PM7/IT4F blend is used, measured with an excitation wavelength at 750 nm, with a fluence of $0.66 \mu\text{J}/\text{cm}^2$.

To verify the reliability of results obtained from the MCR fitting with two components, it is advisable to utilize another TA dataset measured under a different excitation fluence. Since the species derived from TA measurements conducted with different fluences should be the same, one should expect similar or even identical associated spectra for each component from the two independent measurements. However, the concentration evolution may differ due to variations in excitation fluences. If there is a hidden species involved but not accounted for during the MCR decomposition, this hidden species should contribute to a change in the shape of the associated spectra when the TA measurement is done using a different fluence. In other words, if indeed a hidden species is present, the associated spectra from the MCR decomposition of two sets of TA data measured at different excitation fluences should exhibit differences.

However, as shown below in Figure R7, from two TA spectra of the PM7/IT4F blend, measured with two different excitation fluences and analyzed by MCR using two components, we can obtain almost identical associated spectra. The concentration evolution for these two species shows different kinetic behaviors because the initial exciton concentrations are different due to different excitation fluences. This indicates that it is reasonable to decompose the TA spectrum of PM7/IT4F with MCR using two components.

Figure R7. Comparison of the associated spectra and concentration evolution of the components obtained from MCR analysis on the experimental TA spectra of PM7/IT4F, measured with different excitation fluences and an excitation wavelength of 750 nm.

8. One might expect that the fit to 2 species (pre- and post- charge transfer) should be largely sufficient to describe the observed TA data. This can be seen by the fact that two species provide dominant contributions to the TA spectrum in SI Figure 4. What happens if the authors instead limit their deconvolution to 2 species as opposed to 10? Does the error in the fit significantly increase then? And what does a 2D plot of the residual of the extraction as a function of both wavelength and time look like now?

Our response: As clarified in our detailed response to comment 7 from the reviewer, we used only 2 components for the MCR analysis, not 10. Nevertheless, in the revised manuscript, we adhered to the TA analysis protocol outlined in the reference (Energy Environ. Sci., 2022,15, 1545), following the suggestion made in comment 9 from the reviewer.

9. To avoid any questions about the deconvolution altogether, the authors could also instead just use timescales taken from the raw TA data kinetics, which by definition will always be more trustworthy than any sort of spectral deconvolution. For example, PBDB-T family donor polymers have a fairly broad hole polaron PIA that extends beyond the spectral features of the NFA in the TA on the blend. An example of this can be seen in Figure 3b in this paper: 10.1039/D1EE03565G. Here, a kinetic taken from ~1020 nm would yield a kinetic that accurately tracks the rise of the donor polymer hole polaron PIA (i.e. the conversion of S1 into CT states/charges) without any significant overlap with the other spectral features in the blend. Thus, a similar analysis in the authors' work could provide an alternative way to determine the S1 to CT conversion timescale without having to rely on spectral deconvolution and the associated issues.

Our response: In the revised manuscript, we tried to adhere to the TA analysis protocol outlined in the reference (Energy Environ. Sci., 2022,15, 1545), following the suggestion from the reviewer. Here, we analyze only the visible region of the TA spectra, because of the limitation of our TA setup with the generation of white probe light limited to the wavelength range of 400-950 nm for the TA measurements. As demonstrated below in Figure R8 and in Supplementary Figure 4 of the revised manuscript, in the binary blend of PM7/IT4F, we derived a decay time constant of 8 ps from the GSB signal of the donor material at 630 nm. Consistent with the findings in the reference, this signal, resulting from charge carrier generation, persisted for a relatively extended

duration. Then, based on the GSB signals of the donor material, we conclude that, for PM7/IT4F, the S1 to CT conversion time is approximately 8 ps. Also, we observed the GSB signal of PM7 quickly after photoexcitation, which implies an ultra-fast hole transfer at the acceptor/donor interfaces, corresponding to a high k_{DS} value, thanks to the high ΔE_{CT} of the PM7/IT4F blend.

For the binary blend of PM7/Y5, we observed negative signals in the 650-850 nm region (Figure R8 and Supplementary Figure 4), assigned to the GSB of Y5 (steady-state absorption spectra of Y5 shown in Supplementary Figure 3). However, unlike the PM7/IT4F blend, GSB signals from the donor PM7 are hardly observed in the fs-TA spectra for the PM7/Y5 blend. In fact, the TA spectra for the PM7/Y5 system closely resemble those for the neat Y5 film, as shown in Supplementary Figure 5. Moreover, a comparison of the GSB signal for Y5 in the neat film and that for the PM7/Y5 blend reveals very similar decay lifetimes (≈ 140 ps). These experimental TA features indicate that in the PM7/Y5 system, the intrinsic kinetics of Y5 dominate the photophysical process, while the contribution of the PM7 GSB signal is minimal. This could be due to either no dissociation of acceptor excitons (no transfer of holes from Y5 to PM7), or the holes, after transferring from Y5 to PM7, could quickly return from PM7 to Y5, owing to the extremely lower ΔE_{CT} . In both cases, the hole transfer rate is severely limited, and we believe the latter is a more plausible reason, as evidenced by the fact that the PM7/Y5 solar cell still operates with a peak $E_{QE_{PV}}$ close to 10% (Figure 4a of the main text). Therefore, we conclude that the k_{BK} of the PM7/Y5 blend, with a lower ΔE_{CT} , is higher compared to that of PM7/IT4F, corresponding to a lower k_{DS} for the PM7/Y5 blend. Similar results have been reported for other organic donor/acceptor systems, where a reduced ΔE_{CT} leads to a reduction in hole transfer rate by over two orders of magnitudes (ACS Energy Lett. 2021, 6, 8, 2971). Accordingly, we have corrected the discussion regarding the TA results in the revised manuscript.

Figure R8. TA spectra (left) and time evolution of TA signals recorded at the 630 nm and 680 nm (right), for the blends of PM7/IT4F (upper) and PM7/Y5 (lower), excited at 750 nm.

10. To examine the CT state generation efficiency of different acceptors, the authors then compare the S1 to CT conversion timescales from the TA (which are themselves already flawed due to the excessively high signal; see above) with the measured PL lifetime by TCSPC. Whilst I understand the logic behind this, unfortunately the complicated behaviour of the Y-series NFAs means this is not a valid strategy. Specifically, the measured excited state lifetime of Y-series NFA by TA (~300 ps) is typically much shorter than the measured PL lifetimes (~1 ns). Whilst the reason for this discrepancy is unclear (and likely deserving of a paper by itself), I would recommend that the authors measure the TA of the neat acceptor films (again with a fluence series to determine the excited state lifetime in the linear regime), and then use this TA lifetime instead of the PL lifetime in their CT generation efficiency analysis.

Our response: For the revised manuscript (Supplementary Figure 5), we performed TA measurements for the neat acceptor films, using a low pump fluence, and determined the S1 state lifetime of IT4F and Y5, which are 40 and 140 ps, respectively. While the values are considerably lower than that determined from transient PL, the conclusion that IT4F has a slightly shorter excited state life time compared to Y5, remain valid. For the revised manuscript, we replace the transient PL data with the TA data.

11. It would be good to see some TA measurements performed on the three-component systems with different wt% of the second acceptor to confirm what the authors propose. Specifically, that the addition of a small amount of the lower offset NFA (for example, Y5 in the PM7:IT4F system) does indeed result in a small increase in the S1 to CT conversion timescale that at first has minimal impact on the efficiency of this process, but eventually results in a reduction of the charge generation yield. As the TA spectrum will become more complication upon the addition of the second acceptor, perhaps it might be necessary to track the hole polaron PIA, as discussed above, instead of relying on the spectral deconvolution.

Our response: As stated earlier, we no longer rely on the spectral deconvolution for the analysis of the TA spectra for the revised manuscript. In supplementary Figure 4 of the revised manuscript, we provided the TA spectrum of the ternary blend of PM7/IT4F/Y5 with a Y5 content of 20%. From the TA spectrum, the GSB of PM7 (630 nm) is resolved, and from the GSB signal of PM7, we estimated a decay time constant of 30 ps. This value is increased compared to that of the binary blend PM7/IT4F (8 ps), indicating that the addition of Y5 in the ternary blend could lead to an increase in the dissociation time of the S1 state.

To address the reviewer's comment regarding the influence of increasing Y5 content on charge generation yield, we conducted electric field dependent PL measurements, now included in Supplementary Note 2 of the revised manuscript, and shown below in Figure R9.

Specifically, for the PM7/IT4F solar cell, the steady-state PL is significantly quenched compared to the PL emission from the S1 state of neat IT4F (Figure R9a). This suggests a homogeneous mixture between PM7 and IT4F in the blend, allowing for the diffusion of the S1 state to a donor/acceptor interface within the lifetime of the S1 state. It also suggests a fast dissociation of the S1 state at the donor/acceptor interfaces. The PL intensity of the PM7/IT4F solar cell is hardly dependent on the electric field (Figure R9b) because all S1 states can dissociate efficiently without an

electric field. Therefore, the charge generation yield in PM7/IT4F is high, leading to a high EQE_{PV} .

For the PM7/Y5 solar cell, the steady-state PL emission is not significantly quenched compared to that of the neat Y5 film (Figure R9a). This suggests inefficient dissociation of the S1 state in the PM7/Y5 blend. This inefficiency could be attributed either to a too large phase separation between PM7 and Y5, preventing excitons from diffusing to a donor/acceptor interface within the lifetime of S1 states, or to a too low dissociation rate of the S1 states at the donor/acceptor interface. In the first case, the emission is not expected to be electric field-dependent, as the diffusion of the S1 state is not field-dependent. However, electric field-dependent PL measurements for the PM7/Y5 solar cell (Figure R9c) revealed a significant quenching of the PL intensity with increasing electric field. This indicates that the dissociation of the S1 states at PM7/Y5 interfaces is indeed slow but facilitated by the electric field. In fact, the PL signal is quenched by 80% with a sufficiently large applied field, implying that the diffusion of excitons or the morphology of the active layer is not a severe issue for the PM7/Y5 blend. Therefore, the charge generation yield in PM7/Y5 is limited by the dissociation rate of the S1 state, leading to a low EQE_{PV} .

For the ternary PM7/IT4F/Y5 solar cell with a Y5 content of 10%, emission peaks from S1 states of both IT4F and Y5 are observed in the steady-state PL spectra (Figure R9d). Strikingly, we note that the field dependency of both IT4F and Y5 emissions in the ternary blend is stronger than that of the IT4F emission in the PM7/IT4F binary blend. This indicates that the addition of a small amount of Y5 could lead to a reduction in the dissociation rate of the S1 state in the ternary blend. However, this reduction in dissociation rate does not notably affect the charge generation yield, as confirmed by the steady-state PL measurements: The quenching efficiency of the ternary blend with a Y5 content of 10% is similar to that of the binary PM7/IT4F blend (Figure R9a). Therefore, the EQE_{PV} of the ternary solar cell with a Y5 content of 10% is high, close to that of the binary PM7/IT4F solar cell.

For the ternary PM7/IT4F/Y5 solar cell with an 80% Y5 content, the steady-state PL emission is not significantly quenched compared to that of the neat Y5 film (Figure R9a), similar to what was observed for the binary PM7/Y5 solar cell. This suggests an inefficient dissociation of the S1 state in the ternary PM7/IT4F/Y5 solar cell with an 80% Y5 content. Electric field-dependent PL measurements for the ternary PM7/IT4F/Y5 solar cell with an 80% Y5 content (Figure R9e) revealed a significant quenching of the PL intensity with increasing electric field, indicating that the inefficient dissociation of the S1 state originates from the slow dissociation of the S1 states at donor/acceptor interfaces. Therefore, the charge generation yield in the ternary PM7/IT4F/Y5 solar cell with an 80% Y5 content is limited by the dissociation rate of the S1 state, leading to a low EQE_{PV} .

Figure R9. a) Steady state PL spectra of the PM7/IT4F and PM7/Y5 binary blends, the IT4F and Y5 pure films, and the ternary PM7/IT4F/Y5 (10%) and PM7/IT4F/Y5 (80%) films. Electric field dependent PL spectra of the solar cells based on b) PM7/IT4F, c) PM7/Y5, d) PM7/IT4F/Y5 (10%), and e) PM7/IT4F/Y5 (80%). f) PL peak intensity as a function of applied voltage for solar cells based on PM7/IT4F, PM7/Y5, PM7/IT4F/Y5 (10%), and PM7/IT4F/Y5 (80%).

12. In addition, I would be curious to learn more about the dynamics of the two electron acceptor materials in the three-component blend. For example, if the film was excited at something like 580 nm to preferentially pump the PM7 donor, one might expect the initial electron transfer step to largely proceed from PM7 to IT4F, due to the excess wt% of this component vs Y5 in the highest performing system. However, do the authors see any evidence for a delayed increase in the population of the Y5 component (either the GSB or the S1 PIA), potentially indicating some sort of equilibration forming between the PM7:IT4F CT state and the Y5 S1, as suggested by the authors?

Our response: We appreciate the reviewer for suggesting a method to evaluate the dynamics of the excited states in the ternary blends. We have performed fs-TA measurements for the ternary blend of PM7/IT4F/Y5 with a Y5 content of 20%, using an excitation wavelength of 580 nm. As shown below in Figure R10, we observe that there is a broad GSB signal of IT4F with the peak at ~ 700 nm, overlapping with the GSB of Y5. Although the GSB peak positions for these two acceptors must be different,

it is difficult to resolve the GSB of Y5 since its content is low here. The excited-state absorption signals of IT4F and Y5 are in the range of 850-900 nm, also overlapping with each other. Therefore, we are unable to draw a meaningful conclusion regarding the Y5 kinetics via monitoring its GSB or excited-state absorption when exciting the ternary blend at 580 nm.

Figure R10. TA spectrum for the blend of PM7/IT4F/Y5 (20%), excited at 580 nm with a fluence of $0.23 \mu\text{J}/\text{cm}^2$.

13. Whilst the device performance changes in PM7:IT4F:Y5 are quite convincing, I think the situation isn't as clear cut in D18:EC9:SM16. For example, there is only a small reduction in the V_{nr} (0.12 V) from 0.210 to 0.198 V with 10 wt% of SM16, yet the V_{oc} increase is much larger (0.37 V) from 0.869 to 0.906 V. In addition, there is also a noticeable improvement in the FF from 74.9% to 78.5%. Both of these factors, which seem to be separate from the mechanism involving a reduced V_{nr} proposed by the authors, contribute much more significantly to the improved PCE from 17.4% to 19.2%. Thus, I would caution the authors about ascribing the performance improvement seen here, as they could instead be attributed to morphological effects.

Our response: Indeed, there could be additional reasons for the increase in V_{oc} and the overall performance of the solar cells based on D18/EC9/SM16. While a morphological improvement may not be clearly observed from AFM and TEM, it could still contribute to the enhanced performance. Therefore, we have revised the text in section 2.4 of the manuscript to improve clarification, as follow,

“It is noted that the degree of reduction in overall voltage loss exceeds that anticipated from the increase in E_{QEEL} , suggesting that there could be additional reasons for the increase in V_{oc} with the addition of SM16. Nevertheless, neither the peak $E_{QE_{PV}}$ (Figure 5b) nor the FF (Table 2) of the ternary solar cell is significantly reduced by the increased SM16 content, even with a high SM16 content of 20%. This is attributed to the similar molecular structures of SM16 and EC9, leading to similar morphological properties of the active layers, regardless of the SM16 content (Supplementary Figure 15).”

14. Towards the end of the manuscript, the authors also propose that a similar third acceptor strategy should also work in fullerene systems. Whilst I accept that fullerene acceptor OSCs are no longer state of the art, I think if the authors could also demonstrate that their strategy could indeed work in a large offset fullerene blend, it would be a very convincing model system. Thus, I would ask if the authors could try to replicate their

strategy in a fullerene system as further evidence for its effectiveness. This would also go some way to allaying my concerns about drawing too strong conclusions from the D18:EC9:SM16 system, which I find to be a less convincing example of their proposed strategy.

Our response: For the revised manuscript, we verified the effectiveness of the dual-acceptor strategy proposed in this work using a D/A1 system based on a fullerene acceptor, and SM16 as A2. Our strategy was found to be effective for the fullerene system as well, as discussed below and in Supplementary Note 5 of the revised manuscript.

“We have additionally employed SM16 (chemical structure shown in Supplementary Figure 1) as the secondary acceptor for the PM6/PC₇₁BM blend to evaluate the effectiveness of the dual acceptor strategy for improving the performance of OSCs. From the EL spectrum of the PM6/SM16 binary solar cell, we only resolve the emission from the S1 state of SM16 (Supplementary Figure 27a), implying that ΔE_{CT} of the PM6/SM16 blend is low, and k_{DS} is also low. Because of the low k_{DS} , exciton dissociation efficiency in the PM6/SM16 binary solar cell is severely limited, as confirmed by the PL measurements (Supplementary Figure 27b). Accordingly, IQE of the PM6/SM16 binary solar cell is low, leading to low peak EQE_{PV} , about 10% (Supplementary Figure 27c). The S1 state energy of SM16 (1.50 eV, Supplementary Figure 27d) is comparable to the E_{CT} of the PM6/PC₇₁BM blend (1.57 eV, Supplementary Figure 27e). Furthermore, the emission efficiencies of the S1 states of PC₇₁BM (0.01%) is lower than that of SM16 (0.9%) (Supplementary Figure 27f). Accordingly, in the PM6/PC₇₁BM/SM16 ternary blend, k_{DS} is expected to decrease with the increasing SM16 content.

Because of the decrease in k_{DS} , the dominant peak in the EL spectrum (Supplementary Figure 27a) of the PM6/PC₇₁BM/SM16 ternary solar cell changes from the PM6/PC₇₁BM CT state emission peak to the SM16 S1 state emission peak, with the increasing SM16 content, and the device EQE_{EL} significantly increases, as shown in Supplementary Figure 28a: EQE_{EL} of the ternary solar cell with a SM16 content of 10% is increased by over an order of magnitude, compared to that of the PM6/PC₇₁BM binary solar cell. Accordingly, the lifetime of charge carriers in the PM6/PC₇₁BM/SM16 ternary solar cell is increased with the increased SM16 content (Supplementary Figure 28b), and V_{OC} is also increased, as shown in Supplementary Figure 28a. Meanwhile, EQE_{PV} does not decrease with the addition of a small amount of SM16: The peak EQE_{PV} of the solar cell with a SM16 content of 10% is close to that of the PM6/PC₇₁BM binary solar cell (Supplementary Figure 27c). Since the FF of the ternary solar cell is not reduced by the addition of a small amount of SM16 (Supplementary Table 2), PCE of the solar cell is increased from 6.48% to 7.81%, when the SM16 content is increased from 0% to 20% (Supplementary Figure 28c). The basic photovoltaic performance parameters of the solar cells with different SM16 content are summarized in Supplementary Table 2, and the J-V curves are provided in Supplementary Figure 28d.”

Reviewer #2 (Remarks to the Author):

The paper demonstrates that the device electroluminescence quantum efficiency, a critical factor in mitigating non-radiative voltage losses and further improving the

performance of organic solar cells (OSCs), is influenced by the dissociation rate constant of singlet states. This is not the only parameter in mitigating non-radiative voltage losses, so that controlling such parameter will not warrant a substantial rise in efficiency for OSCs. The paper indicates an experimental design which would push the limit, but does not clearly quantify its impact. The paper is well written, deserves to be published, but certainly lacks the urgency and broad impact needed for a Nature communications paper. I suggest to reject it.

Our response: We appreciate the reviewer's efforts in providing feedback on our manuscript. Indeed, the non-radiative decay of excited states in organic solar cells is always associated with several parameters. In this work, we analytically and experimentally demonstrate a new strategy, focusing on the dissociation rate constant of the singlet excitons, which has not been seriously considered thus far. This approach aims to overcome the voltage bottleneck and enhance the overall performance of organic solar cells. We validated the proposed strategy in this work using several classic active material systems. Consequently, we believe that the results presented here will contribute to the accelerated development of more efficient materials and more effective device engineering strategies, ultimately improving the *PCE* of organic solar cells. We anticipate that these findings will positively impact the field of organic solar cells, as well as organic electronics and photovoltaic technology on a broader scale.

Reviewer #3 (Remarks to the Author):

The authors present a hypothesis for OPV solar cells, which suggests that slowing down the charge transfer rate (from S1 to CT state) to a certain extent can be beneficial for power conversion efficiency, as it enhances EQE_EL without compromising current generation. They base this hypothesis on a simulation of the PCE as a function of the rates of different CT and back transfer processes. They experimentally support their hypothesis by testing three ternary OPV blends containing two acceptors, which they claim reduces the CT rate. The story could in principle be of interest, even for a high impact journal such as Nat. Commun., but it cannot be accepted currently. Especially since the superior performance of ternary blends is not a new concept, they need to prove their hypothesis much more strongly. Currently, there are serious problems with the data, the interpretation and the substantiation of the conclusions.

Our response: We thank the reviewer for the overall positive comments.

- 1) The assumptions made when deriving their hypothesis are too simplistic. The authors say that a low CT-S1 offset leads to slow charge transfer rates. This is not necessarily true when the re-organization energy is low. Many studies show ultrafast charge transfer at very low offsets. Also, they authors claim that the back transfer rate is higher when the forward rate is slower. Again, this is not necessarily true. If the Marcus curve peaks at zero driving force, both rates will be maximized, leading to an equilibrium between the two states. The authors should revise their understanding of Marcus theory.

Our response: We agree with the reviewer, that the transfer rate forward or backward is also strongly dependent on the reorganization energy. Therefore, we revised our manuscript, and clarified that the transfer rate depends not only on the energy offset, but also other factors such as reorganization energy.

More specifically, in section 2.1, we have deleted the statement “...and a decrease in k_{DS} often leads to an increase in k_{BK} (since the energy difference between the S1 and CT states is the most important factor determining the transfer rate, according to the Marcus theory⁴¹)”.

In the last paragraph of section 2.1 of the revised manuscript, we have corrected a statement as “Anticipating that r_{BK} does not always reduce with a reduction in k_{DS} , it follows that an intentional decrease in k_{DS} within real-world OSCs could lead to an elevation in PCE.”

Additionally, we have revised the first paragraph of section 2.2 as “Practically, when considering binary OSCs, reducing k_{DS} could require a reduction in the energy difference between the CT state and S1^A (ΔE_{CT})⁴³. Achieving this requires precise adjustments to the energy levels of the frontier molecular orbitals of either the donor or acceptor material^{44,45}. However, the task of such precise calibration presents challenges. Additionally, k_{DS} depends on reorganization energy and electronic coupling between the S1 and CT states, both of which are even more challenging to adjust. Consequently, there is still a lack of an effective experimental strategy for optimizing the trade-off between IQE and EQE_{EL}, through the regulation of k_{DS} .”

Then, in the third paragraph of section 2.2, we have revised a statement as “Assuming the electronic coupling and reorganization energy of the blend of D/A1 are not significantly different from those of classic donor/acceptor blends used for organic solar cells, this high ΔE_{CT1} is expected to lead to a high dissociation rate constant of S1^{A1} (k_{DS}^{A1}) according to the Marcus theory^{52,53}.”

- 2) The TA measurements cannot be taken seriously, because the fluence used was way too high. To avoid non-linear artefacts in NFA blends, TA signals need to be below 1×10^{-3} OD. Here, they are about 20 times higher.

Our response: We appreciate the reviewer for pointing out the inappropriate use of pump fluence for the TA measurements. For the revised manuscript, we have repeated the TA measurements using different fluences and identified a level low enough to avoid influencing the dynamics of the excited states. Nevertheless, we attempted to refrain from making overly strong conclusions based on the TA analyses and opted to relocate the TA data to the supplementary information.

- 3) The authors claim a S1 decay rate of 800 ps for one of the acceptors, justifying that slower CT rates are not a problem. However, they find this time constant based on nanosecond-resolved PL measurements. They miss out all the fast components of the exciton decay. Typically, the S1 lifetime in films is very multi-exponential with fast components that can very well compete with the CT time.

Our response: Initially, our objective was to utilize transient PL to demonstrate that the acceptors used had similar excited state lifetimes. This would allow us to primarily attribute the increased EQE_{EL} in the ternary blend to the reduced dissociation rate (k_{DS}) of the singlet state. For the revised manuscript, we conducted TA measurements on neat acceptor films based on IT4F and Y5 (Supplementary Figure 5), using low pump fluences, and indeed derived faster lifetime values for the acceptors: 40 ps and 140 ps

for IT4F and Y5, respectively. In the revised manuscript, we replaced the lifetime values determined from transient PL measurements with those from the TA measurements, and the conclusion of the manuscript remains unaffected.

- 4) The structural characterization by AFM (and by using molecules of similar structure) is not at all sufficient. This can have very profound influence on the photophysical processes. For example, the charge transfer rates of 20 ps and 70 ps are quite slow (usually it is on the <500 fs scale for efficient systems). This is likely because there is exciton diffusion through neat donor and acceptor domains before the CT step. The complex morphology of the ternary blends must be understood and controlled in detail in order to exclude this.

Our response: In this work, we used ternary strategy to reduce ΔE_{CT} of organic solar cells, for a reduction in k_{DS} and an increase in $E_{QE_{EL}}$. Indeed, as suggested by the reviewer, alterations in the morphology of the active layer can lead to changes in the stacking structure of donor/acceptor molecules within the active layer. Consequently, this can induce modifications in the electronic structure of the donor and acceptor molecules at their interfaces, subsequently impacting molecular parameters such as E_{CT} , E_{SI} , and reorganization energy, which determine the dynamic properties of the CT state. The changes in these molecular parameters can ultimately attribute additionally to a modification of the k_{DS} .

Figure R11. TEM images of the PM7/IT4F and PM7/Y5 blends. The TEM images indicate that the morphologies of the active layers based on PM7/IT4F and PM7/Y5 are different, and a significant increase in Y5 content in the ternary blend of PM7/IT4F/Y5 is expected to lead to a morphological change of the active layer.

For the revised manuscript, we conducted additional characterization of the binary and ternary systems used in this work using TEM (Supplementary Figure 10, and above in Figure R11). The results align with those from AFM. Taking PM7/IT4F as an example, we observed that the donor and acceptor molecules in the binary PM7/IT4F blend are homogeneously mixed. The addition of a small amount of Y5 has little impact on the morphology of the blend film. However, when the Y5 content exceeds 40%, phase separation in the ternary film increases, and its morphology begins to resemble that of the binary PM7/Y5 film.

The different morphological properties of the blends based on the IT-series acceptors and the Y-series acceptors can also be observed through GIWAXS measurements. As can be seen in Figure R12 below, the GIWAXS pattern of the blend of PM7/IT4F is noticeably different from that of the blend of PM7/Y6, indicating that the molecular structural differences between these acceptors can indeed cause

morphological differences.

Additionally, we have characterized the morphology of the PM7/Y6 blend, using TEM and AFM (Supplementary Figure 7), and compared the results with those of the PM7/Y5 blend. From the AFM and TEM images, we note that the morphology of the PM7/Y5 film is similar to that of the PM7/Y6 film. Given the high efficiency of the solar cell based on PM7/Y6, it is expected that the morphology of the PM7/Y6 blend is optimized for the diffusion of S1 states to a donor/acceptor interface, and similarly, the morphology of the PM7/Y5 blend is expected to be optimized.

Figure R12. 2D GIWAXS patterns of the blend of PM7/IT4F (left) and PM7/Y6 (right).

Therefore, from the morphology characterization, we conclude for the ternary PM7/IT4F/Y5 solar cells that the introduction of a small amount of Y5 has little impact on the morphology of the active layer, while the introduction of a large amount of Y5 does lead to a change in the morphology of the active layer. However, the morphology of the PM7/Y5 blend, as well as that of the ternary PM7/IT4F/Y5 blend, is optimized for the diffusion of excitons to a donor/acceptor interface within their lifetime.

It is highly challenging to determine whether the change in the morphology of the active layer contributes, in addition to the reduced ΔE_{CT} , to the reduction in k_{DS} . Nevertheless, this does not affect our main conclusion that the ternary strategy allows us to achieve a reduction in k_{DS} for an increase in EQE_{EL} , which is the main message we want to convey in the manuscript. For the revised manuscript, we have provided further evidence for the reduced k_{DS} associated with the incorporation of Y5 as a second acceptor in the PM7/IT4F blend, from the electric field-dependent PL measurements, as discussed in supplementary Note 2 of the revised manuscript.

- 5) It is not new that ternary blends perform better and yield high efficiencies. The authors present a new hypothesis for the reason. However, they do not rule out other possible reasons for the improved performance. In particular, the morphology could have an important effect. Also, the charge generation is likely complicated by energy transfer processes between the different blend constituents, and there might be charge hopping between the two acceptors. This needs to be investigated to make sure the hypothesis of the authors is correct.

Our response: In this work, we demonstrate that EQE_{EL} is increased, as a result of reduced k_{DS} , via the ternary strategy. This leads to increased performance of organic

solar cells. The crucial determining factors for EQE_{EL} , apart from k_{DS} , include E_{CT} and reorganization energy, since EQE_{EL} depends exponentially on both of these parameters (PHYSICAL REVIEW B 2010, 81, 125204). Changes in the morphology of the active layer may result in an increase in E_{CT} or a decrease in reorganization energy. Therefore, as the reviewer pointed out, apart from reduced k_{DS} , a morphology improvement induced by the use of a second acceptor could also be a possible reason for the improvement in EQE_{EL} .

For the ternary system studied in this work, the second acceptor introduced has a lower bandgap energy compared to the primary acceptor (Supplementary Figure 3), and after blending with the donor, it exhibits a lower E_{CT} . Thus, the introduction of the second acceptor, even if it induces changes in the morphology of the active layer, should not lead to an increase in E_{CT} .

Figure R13. Sensitive EL and EQE_{PV} spectra of the PM7/IT4F, PM7/IT4F/Y5, and PM7/Y5 solar cells. The E_{CT} and reorganization energy (λ) were determined by Gaussian fitting to the lower energy part of the EQE_{PV} spectrum using the method described in the literature (J. Mater. Chem. A, 2021, 9, 19770).

Additionally, taking the PM7/IT4F/Y5 solar cells as an example, we measured the EL and sensitive EQE_{PV} spectra of the binary solar cell based on PM7/IT4F for the revised manuscript, as shown above in Figure R13 and in Supplementary Figure 6 of the revised manuscript. The reorganization energy was determined to be approximately 0.32 eV through Gaussian fitting to the tail of the EQE_{PV} spectrum. After the introduction of Y5, we observed a slight decrease in reorganization energy. However, even with a Y5 content of 20%, the reorganization energy remained at 0.28 eV. Therefore, the introduction of the second acceptor does not significantly reduce the reorganization energy. Thus, we believe that the introduction of the second acceptor while it can cause changes in the morphology of the active layer, will not enhance

EQE_{EL} by increasing E_{CT} or reducing reorganization energy.

Finally, to address the reviewer's comment on the possible influence of energy transfer processes on the conclusion of the manuscript, we also use the PM7/IT4F/Y5 system as an example. We agree that singlet excitons may transfer directly from IT4F to Y5 or first form a CT state and then transition to the S1 state of Y5. Since the k_{DS} of Y5 is lower than that of IT4F, both possibilities would ultimately result in a lower k_{DS} in the ternary system compared to the binary system of PM7/IT4F. Therefore, the introduction of A2 with a lower k_{DS} is still expected to lead to an improvement in EQE_{EL} of organic solar cells, with or without direct energy transfer between the acceptors, which is the main message that we want to convey with our manuscript. In section 2.3 of the revised manuscript, we have added a discussion for a better clarification.

“It should be noted that the addition of Y5 to the binary blend of PM7/IT4F could lead to a change in the morphology of the active layer, potentially resulting in a reduction in the reorganization energy of the active layer. This, in turn, could contribute additionally to the increase in EQE_{EL} ^{23,24}. Therefore, we measured the EL and sensitive EQE_{PV} spectra of the solar cell based on PM7/IT4F, PM7/IT4F/Y5, and PM7/Y5 (Supplementary Figure 6) and determined the reorganization energy through Gaussian fitting to the tail of the EQE_{PV} spectra. We found that the reorganization energy is approximately 0.32 eV for the PM7/IT4F binary blend, and it slightly decreases to 0.28 eV in the ternary blend with a Y5 content of 20%. This suggests that the introduction of Y5, while it may cause changes in the morphology of the active layer, does not significantly decrease the reorganization energy. Therefore, the increased EQE_{EL} should not be ascribed to a morphological reason.”

REVIEWER COMMENTS

Reviewer #1 (Remarks to the Author):

The authors have clearly dedicated significant effort to revising their manuscript in response to the points raised by the referees, which is appreciated. As a result, the manuscript has been greatly improved and is now nearly ready for publication. I just have some minor remaining points that it would be good if the authors could address before I would be pleased to recommend acceptance of this work.

1. It is great to see the fullerene blend data, and this seems to confirm the authors' hypothesis. One question: in the PM6/PC71BM/SM16 blend with 10-40% SM16, the EQE in the SM16 absorption region around 800 nm is much higher than in the 80% wt% or PM6:SM16 binary blend. This implies that the charge generation from SM16, which is very poor in the binary blend, is greatly enhanced in the presence of PC71BM. Do the authors have an explanation for this interesting observation?

2. On page 11 of the main text, the authors state: '...the time evolution of the TA signals at 630 nm is fitted using the sum of two exponential functions and an instrument response function (IRF), giving rise to a rise time constant of ≈ 8 ps.'

Since the authors used a biexponential function yet only provide a single rise time, it is not clear where the value of ≈ 8 ps has come from. Is it the weighted average of the two time constants from the biexponential fit? If so, averaging out the two time constants obscures a lot of subtleties in the data. For example, the faster time constant may relate to the ultrafast charge transfer from singlet excitons generated near the D/A interface, whilst the longer one could represent the diffusion of singlet excitons to the D/A interface. I would suggest that the authors provide the two time constants (and their respective weightings from the prefactors of the two exponential terms), rather than the single time constant currently provided.

3. On page 12 of the main text, the authors state: 'This could be due to either no dissociation of acceptor excitons (no transfer of holes from Y5 to PM7), or the holes, after transferring from Y5 to PM7, could quickly return from PM7 to Y5, owing to the extremely lower ΔE_{CT} . In both cases, the hole transfer rate is severely limited, and we believe the latter is a more plausible reason, as evidenced by the fact that the PM7/Y5 solar cell still operates with a peak EQEPV close to 10% (Figure 4a).

I think the authors' observations are better explained by the minimal dissociation of acceptor excitons, rather than significant dissociation of acceptor excitons followed by fast CT state recombination. If it was

the latter, I would expect to see a significant (but short lived) GSB of the polymer, which is not present in the TA data.

4. The V_{nr} of the PM7/IT4F device decreases by 85 mV from 0.354 to 0.269 V upon the addition of 10 wt% Y5. However, the V_{oc} only increases by 42 mV from 0.827 to 0.869 V. Do the authors have any suggestion of why the full reduction in V_{nr} from the EQE_EL measurement is not reflected in the V_{oc} ?

5. I wanted to check if the authors have calculated the excitation fluences correctly. For example, in the rebuttal document, they state that an excitation power of 4.5 μW corresponds to a fluence of 0.13 $\mu\text{J}/\text{cm}^2$. However, running the maths through (using the laser rep rate of 1 kHz given in the Methods i.e. excitation at 500 Hz for the pump on and pump off shots) reveals that this corresponds to a pump diameter of $\sim 3000 \mu\text{m}$, assuming a gaussian beam profile. This seems extremely large for a TA measurement; pump diameters will of course vary between experimental setups, but I would have expected a diameter more like 300 μm .

6. Most of the TA kinetics are shown on a linear time axis. However, this makes it difficult to see the kinetics for timescales $< 50 \text{ ps}$. Could the authors instead plot the time axis for their TA kinetics on log scale to better visualise the early time data (like in Fig. R6b).

7. In their analysis on the effect of adding the third component on the EQE_EL and V_{oc} , the authors plot double y-axis graphs showing the EQE_EL and V_{oc} as a function of the third component wt % (for example Fig. 4b, but also repeated many times in the SI for the other blends studied). However, it is not clear at what injection current the EQE_EL has been measured. This could be an important consideration, as the EQE_EL seems to be strongly dependent on the injection current, at least for the neat film devices for which the EQE_EL curves are given (e.g. Fig. S9 and others). It is important that the authors show the EQE_EL curves for all the blend devices studied in the SI along with the double y-axis plots so that the reader can access the data themselves. Further, can the authors clarify at which injection current the EQE_EL value is taken from? Best practice is that the EQE_EL should be taken at injected current densities roughly equivalent to J_{sc} (here $\sim 20 \text{ mA}/\text{cm}^2$).

8. Singlet excitons are referred to by the abbreviation 'S1' throughout. It would be good if the '1' could be in subscript.

9. The authors should be careful with their rounding, as they state that the V_{nr} of the PM7:IT4F:Y5 device is 0.26 V in the text, but Table 1 gives a value of 0.269 V, which should be rounded to 0.27 V.

10. As a suggestion: if the authors want to improve the clarity of their TA spectra at the new lower fluences measured, they could apply moving average smoothing to their data. A gentle moving average smoothing of say 5 adjacent spectral datapoints could remove some of the noise whilst not impacting the integrity of the data. But this is at the authors' discretion and not essential.

Reviewer #2 (Remarks to the Author):

I appreciate the efforts of the authors. The paper has improved, but I cannot change my suggestion to reject, in the sense that the results and the concepts behind do not have the outstanding quality required by Nature Communications. They do not demonstrate immediately, either experimentally, or through a novel strategy, a way to improve OPV's results. It remains a fine paper, but my suggestion is to resubmit elsewhere.

Reviewer #3 (Remarks to the Author):

The authors have done a very thorough job in revising the manuscript, adding additional data (e.g. low fluence TA) and solidifying the analysis. This is now less dependent on MCR and more linked to real TA dynamics. Overall, the concerns of Reviewer 1 and myself are well addressed. I believe that the novelty is sufficient for Nature Commun (unlike Referee 2), and think that the manuscript can now be accepted.

Reviewer #1 (Remarks to the Author):

The authors have clearly dedicated significant effort to revising their manuscript in response to the points raised by the referees, which is appreciated. As a result, the manuscript has been greatly improved and is now nearly ready for publication. I just have some minor remaining points that it would be good if the authors could address before I would be pleased to recommend acceptance of this work.

Our response: We appreciate the positive comments from the reviewer.

1. It is great to see the fullerene blend data, and this seems to confirm the authors' hypothesis. One question: in the PM6/PC71BM/SM16 blend with 10-40% SM16, the EQE in the SM16 absorption region around 800 nm is much higher than in the 80% wt% or PM6:SM16 binary blend. This implies that the charge generation from SM16, which is very poor in the binary blend, is greatly enhanced in the presence of PC71BM. Do the authors have an explanation for this interesting observation?

Our response: Indeed, the EQE of the binary solar cell based on PM6/SM16 is observed to be very low, attributed to the extremely small energy offset between the singlet and the CT state. In this scenario, the exciton dissociation constant (k_{DS}) is very low, thereby limiting charge generation efficiency. In the ternary solar cell based on PM6/PCBM/SM16, one might anticipate the formation of two CT states with different electronic properties at the PM6/PCBM and PM6/SM16 interfaces. Then, the energy offset at the PM6/PCBM interfaces should be high, leading to highly efficient charge generation, while the offset at the PM6/SM16 interfaces is low, akin to the binary PM6/SM16 solar cell, resulting in limited charge generation efficiency at the PM6/SM16 interfaces. Accordingly, EQE signals of the ternary device in the long-wavelength range, attributed to SM16 absorption, are expected to remain low despite the presence of PCBM. Additionally, the ternary device could be represented by a simple parallel connection of PM6/PCBM and PM6/SM16 devices, and the heterojunction system with lower V_{oc} would limit V_{oc} of the ternary device.

Contrary to expectations, we observe an increase in V_{oc} values with the rising SM16 content in the ternary device, accompanied by a significant enhancement in EQE. This improvement in EQE is notable when compared to the binary PM6/SM16 solar cell, particularly in the wavelength region around 800 nm, corresponding to SM16 absorption. This observation suggests the formation of a new effective medium, representing an averaged electronic property of the CT states in the ternary system (J. Phys. Chem. Lett. 2016, 7, 3936). In this scenario, the effective k_{DS} of the ternary system is expected to lie between that of the binary PM6/PCBM and PM6/SM16 systems. Due to the higher k_{DS} of the binary PM6/PCBM compared to that of PM6/SM16, the k_{DS} of the ternary blend are expected to increase with rising PCBM content. Consequently, the charge generation efficiency from SM16 in the ternary system is enhanced, especially when the PCBM content in the ternary blend is high.

2. On page 11 of the main text, the authors state: ‘...the time evolution of the TA signals at 630 nm is fitted using the sum of two exponential functions and an instrument response function (IRF), giving rise to a rise time constant of ≈ 8 ps.’

Since the authors used a biexponential function yet only provide a single rise time, it is not clear where the value of ≈ 8 ps has come from. Is it the weighted average of the two time constants from the bi-exponential fit? If so, averaging out the two time constants obscures a lot of subtleties in the data. For example, the faster time constant may relate to the ultrafast charge transfer from singlet excitons generated near the D/A interface, whilst the longer one could represent the diffusion of singlet excitons to the D/A interface. I would suggest that the authors provide the two time constants (and their respective weightings from the prefactors of the two exponential terms), rather than the single time constant currently provided.

Our response: Indeed, two time constants are derived from the biexponential fit, and the rise time of 8 ps mentioned in the manuscript corresponds to the faster time constant, not an average of the two. In the revised manuscript, we have included the fit parameters used to determine the rise time constants.

3. On page 12 of the main text, the authors state: ‘This could be due to either no dissociation of acceptor excitons (no transfer of holes from Y5 to PM7), or the holes, after transferring from Y5 to PM7, could quickly return from PM7 to Y5, owing to the extremely lower ΔE_{CT} . In both cases, the hole transfer rate is severely limited, and we believe the latter is a more plausible reason, as evidenced by the fact that the PM7/Y5 solar cell still operates with a peak EQEPV close to 10% (Figure 4a).

I think the authors’ observations are better explained by the minimal dissociation of acceptor excitons, rather than significant dissociation of acceptor excitons followed by fast CT state recombination. If it was the latter, I would expect to see a significant (but short lived) GSB of the polymer, which is not present in the TA data.

Our response: We speculated that minimal dissociation of acceptor excitons was not the main reason, given that we managed to obtain a peak EQE of 10%. However, it is indeed arguable whether a 10% peak EQE is sufficiently high to explain the minimal contribution of the PM7 GSB signal. Therefore, for better clarification, we have removed the statement “we believe the latter is a more plausible reason, as evidenced by the fact that the PM7/Y5 solar cell still operates with a peak EQEPV close to 10% (Figure 4a)” from the manuscript.

4. The V_{nr} of the PM7/IT4F device decreases by 85 mV from 0.354 to 0.269 V upon the addition of 10 wt% Y5. However, the V_{oc} only increases by 42 mV from 0.827 to 0.869 V. Do the authors have any suggestion of why the full reduction in V_{nr} from the EQE_{EL} measurement is not reflected in the V_{oc} ?

Our response: For organic solar cells, V_{oc} is influenced not only by V_{nr} but also by E_{CT} and the radiative voltage loss (V_r). While our primary focus in this work is on reducing V_{nr} through the dual-acceptor strategy, the introduction of the second acceptor could lead to minor changes in both E_{CT} and V_r . These changes may contribute to an additional reduction in overall voltage losses. Furthermore, it is noteworthy that the variation in the V_{oc} values of organic solar cells, based on our experience, can be as high as 20 mV, as indicated in the tables for the PV parameters. Additionally, the variations in V_{nr} values determined from the EQEEL measurements are estimated to be around 5-10 mV. Nevertheless, we emphasize that the observed trend of V_{nr} reduction with the increasing content of the second acceptor is consistently clear and highly reproducible.

5. I wanted to check if the authors have calculated the excitation fluences correctly. For example, in the rebuttal document, they state that an excitation power of 4.5 μ W corresponds to a fluence of 0.13 μ J/cm². However, running the maths through (using the laser rep rate of 1 kHz given in the Methods i.e. excitation at 500 Hz for the pump on and pump off shots) reveals that this corresponds to a pump diameter of \sim 3000 μ m, assuming a gaussian beam profile. This seems extremely large for a TA measurement; pump diameters will of course vary between experimental setups, but I would have expected a diameter more like 300 μ m.

Our response: We appreciate the reviewer's comment. Upon careful review, we re-measured the laser beam sizes, identifying an error in the determination of their diameter, likely caused by the displacement of the low distortion faceplate. The accurate beam diameter is 594 μ m at an excitation power of 4.5 μ W. Consequently, the corrected excitation fluence, measured in μ J cm⁻², is 1.7 μ J cm⁻². This correction has been incorporated into the revised manuscript. We extend our gratitude to the reviewer for bringing this matter to our attention and value their diligence in the review process.

6. Most of the TA kinetics are shown on a linear time axis. However, this makes it difficult to see the kinetics for timescales $<$ 50 ps. Could the authors instead plot the time axis for their TA kinetics on log scale to better visualise the early time data (like in Fig. R6b).

Our response: In the revised manuscript, we have plotted the time axis for the TA kinetics on log scale.

7. In their analysis on the effect of adding the third component on the EQE_{EL} and V_{oc} , the authors plot double y-axis graphs showing the EQE_{EL} and V_{oc} as a function of the third component wt % (for example Fig. 4b, but also repeated many times in the SI for the other blends studied). However, it is not clear at what injection current the EQE_{EL} has been measured. This could be an important consideration, as the EQE_{EL} seems to be strongly dependent on the injection current, at least for the neat film devices for which the EQE_{EL} curves are given (e.g. Fig. S9 and others). It is important that

the authors show the EQE_EL curves for all the blend devices studied in the SI along with the double y-axis plots so that the reader can access the data themselves. Further, can the authors clarify at which injection current the EQE_EL value is taken from? Best practice is that the EQE_EL should be taken at injected current densities roughly equivalent to J_{sc} (here ~ 20 mA/cm²).

Our response: The EQEEL values are determined using an injection current density equal to the J_{sc} of the device. This is now clarified in the revised manuscript. Furthermore, the EQEEL curves for the solar cells studied in this work are provided in the supplementary information of the revised manuscript.

8. Singlet excitons are referred to by the abbreviation 'S1' throughout. It would be good if the '1' could be in subscript.

Our response: The '1' is now in subscript in the revised manuscript.

9. The authors should be careful with their rounding, as they state that the V_{nr} of the PM7:IT4F:Y5 device is 0.26 V in the text, but Table 1 gives a value of 0.269 V, which should be rounded to 0.27 V.

Our response: We thank the reviewer for pointing out the error, which is now corrected in the revised manuscript.

10. As a suggestion: if the authors want to improve the clarity of their TA spectra at the new lower fluences measured, they could apply moving average smoothing to their data. A gentle moving average smoothing of say 5 adjacent spectral datapoints could remove some of the noise whilst not impacting the integrity of the data. But this is at the authors' discretion and not essential.

Our response: A gentle moving average smoothing, employing 5 adjacent data points, is applied to the TA spectra in the revised manuscript.

Reviewer #2 (Remarks to the Author):

I appreciate the efforts of the authors. The paper has improved, but I cannot change my suggestion to reject, in the sense that the results and the concepts behind do not have the outstanding quality required by Nature Communications. They do not demonstrate immediately, either experimentally, or through a novel strategy, a way to improve OPV's results. It remains a fine paper, but my suggestion is to resubmit elsewhere.

Our response: In this work, we not only achieved improved OPV performance experimentally but also theoretically, employing a method focused on reducing the dissociation rate of singlet states. Most importantly, this accomplishment involved mitigating non-radiative voltage losses, a bottleneck issue in the field of OPV.

Furthermore, the approach proposed in this work does represent a novel strategy that has not been reported before. Therefore, we believe that our manuscript is highly suited for the prestigious journal Nat. Commun.

Reviewer #3 (Remarks to the Author):

The authors have done a very thorough job in revising the manuscript, adding additional data (e.g. low fluence TA) and solidifying the analysis. This is now less dependent on MCR and more linked to real TA dynamics. Overall, the concerns of Reviewer 1 and myself are well addressed. I believe that the novelty is sufficient for Nature Commun (unlike Referee 2), and think that the manuscript can now be accepted.

Our response: We appreciate the positive comments from the reviewer.

REVIEWERS' COMMENTS

Reviewer #1 (Remarks to the Author):

The authors have responded well to the remaining comments and I'd be very pleased to recommend acceptance. I'd like to extend my congratulations to the authors for their very nice paper!

One very minor thing: in the caption of Figure S4, the faster time constant of the fit is given as 896 ps. Should this be fs?

Reviewer #1 (Remarks to the Author):

The authors have responded well to the remaining comments and I'd be very pleased to recommend acceptance. I'd like to extend my congratulations to the authors for their very nice paper!

One very minor thing: in the caption of Figure S4, the faster time constant of the fit is given as 896 ps. Should this be fs?

Our response: Thank you very much for your positive feedback and recommendation for acceptance. We appreciate your kind congratulations on our paper. Regarding your query about the caption of Figure S4, we have re-evaluated the information, and the time constant of the fit is indeed correctly represented as 896 ps, not fs.